# Engineering the stambomycin modular polyketide synthase yields 37-membered mini-stambomycins

Li Su [1,2,5], Laurence Hôtel[2], Cédric Paris[3], Clara Chepkirui[4], Alexander O. Brachmann[4], Jörn Piel[4], Christophe Jacob [1✉], Bertrand Aigle [2✉] & Kira J. Weissman [1✉]

The modular organization of the type I polyketide synthases (PKSs) would seem propitious for rational engineering of desirable analogous. However, despite decades of efforts, such experiments remain largely inefficient. Here, we combine multiple, state-of-the-art approaches to reprogram the stambomycin PKS by deleting seven internal modules. One system produces the target 37-membered mini-stambomycin metabolites — a reduction in chain length of 14 carbons relative to the 51-membered parental compounds — but also substantial quantities of shunt metabolites. Our data also support an unprecedented off-loading mechanism of such stalled intermediates involving the C-terminal thioesterase domain of the PKS. The mini-stambomycin yields are reduced relative to wild type, likely reflecting the poor tolerance of the modules downstream of the modified interfaces to the non-native substrates. Overall, we identify factors contributing to the productivity of engineered whole assembly lines, but our findings also highlight the need for further research to increase production titers.

[1] Université de Lorraine, CNRS, IMoPA, F-54000 Nancy, France. [2] Université de Lorraine, INRAE, DynAMic, F-54000 Nancy, France. [3] Université de Lorraine, LIBio, F-54000 Nancy, France. [4] Institute of Microbiology, Eidgenössische Technische Hochschule (ETH) Zurich, 8093 Zurich, Switzerland. [5]Present address: Max-Planck-Institute for Terrestrial Microbiology, Department of Natural Products in Organismic Interactions, 35043 Marburg, Germany. ✉email: christophe.jacob@univ-lorraine.fr; bertrand.aigle@univ-lorraine.fr; kira.weissman@univ-lorraine.fr

For almost 30 years, efforts have been made to leverage the modular genetic architecture of the type I polyketide synthases (PKSs) to generate novel derivatives, typically by modifying individual catalytic domains. Despite enormous progress in establishing domain structure–function relationships[1,2], such genetic manipulation remains inefficient[3]. Initial sight into factors potentially contributing to low product yields was provided by cryo-electron microscopy analysis of a model PKS module at multiple stages of its catalytic cycle[4,5]. This work revealed that interdomain contacts are critical for establishing the various functional states of the module, and that transitions between such states rely on evolving interfaces between the domains, as well as the intervening 'linker' regions. This view of modular function was recently reinforced by cryo-EM/crystallographic analysis of additional modules sourced from two PKS systems[6,7]. In short, PKS modules appear to be highly integrated units, thus explaining why exchange of catalytic domains for heterologous counterparts is often detrimental[8]. Collectively, these observations motivate future approaches in which modules or multi-modular subunits are employed as the basic building blocks for engineering the assembly lines[9–15]. Such strategies could be exploited to engineer chimeras between multiple PKS systems, or to generate internally truncated forms of single PKSs, providing access to structurally-simplified analogues (minimal pharmacophores[16]) for biological evaluation.

Nevertheless, using modules requires a clear definition of their domain composition. Classically, modules encompass the three invariable domains required for monomer selection and chain extension (ketosynthase (KS), acyl transferase (AT) and acyl carrier protein (ACP)), as well as any intervening β-keto processing activities (e.g., ketoreductase (KR), dehydratase (DH), and enoyl reductase (ER)) (Fig. 1a), and thus have functional meaning. However, a domain set potentially more relevant to genetic engineering was recently suggested by the finding that KS domains in certain PKSs co-evolve with the tailoring domains located upstream in the assembly lines[17,18]. To avoid confusion with alternative module definitions, we suggest that the term eXchange Unit (XU) that is used in the nonribosomal peptide synthetase (NRPS) field[19] be adopted for this set of domains. Accordingly, a PKS XU begins with the modifying domains and the associated AT, and terminates with the KS that is assigned to the downstream functional module (Fig. 2). Even before these alternative domain sets were identified, engineering efforts revealed that maintaining the key $ACP_n/KS_{n+1}$ interface can, in certain cases, be critical for the function of a hybrid PKS[9,12,20].

Recently, we engineered hybrid PKSs based on both of these domain sets, by covalently tethering heterologous modules to a common donor module within a bimodular mini-PKS[21,22]. Overall, our data demonstrated that the use of both classical module boundaries and XUs led to functional hybrid PKSs, and which domain sets worked best depended on the source module[22]. Indeed, regardless of which extremities are employed, module exchange results in non-native interdomain interactions ($ACP_n/KS_{n+1}$ or $KS_{n+1}/ACP_{n+1}$), and in the case of classical module boundaries, potential incompatibilities in terms of KS substrate specificity (Fig. 2)—both of which have been shown to reduce activity via detailed studies in vitro[23–25].

An alternative to the covalent fusion approach is to create alternative junctions between modules located on distinct subunits[3]. In this case, chain transfer not only depends on the employed modules, but also on the presence of compatible protein–protein interaction motifs called docking domains (DDs)[26] situated at the extreme C- and N-termini of the subunits (Figs. 1 and 2). In native PKSs, matched pairs of such DDs form specific complexes at intersubunit interfaces, enforcing a strict subunit ordering within the systems. To date, multiple mini-PKSs

have been engineered based on non-native module combinations and suitable DDs. However, as much of this effort centered on the erythromycin (DEBS) and related macrolide PKSs, and both classical modules[11,27,28] and XUs[13,20] functioned in these contexts, the applicability of these findings to other PKS systems is not clear. Furthermore, the limited work that has been carried out on intact assembly lines[14,29–32] has focused on engineering hybrid systems.

Here, we leverage subunit-based engineered to substantially shorten the PKS responsible for biosynthesis of the anti-cancer stambomycins 1 in Streptomyces ambofaciens ATCC23877[33]. Using both the classical module definitions and PKS XUs, we generate 37-membered derivatives of the normally 51-membered macrolides (Fig. 1b), albeit in moderate yields. The identified derivatives also clarify the relative timing of the two cytochrome P450-catalyzed hydroxylations at C-28 and C-50 (Fig. 1b). Furthermore, we report an example of intersubunit crosstalk resulting in thioesterase (TE)-mediated release of shunt metabolites. Taken together with recent work by others[12], our data establish guidelines for module/subunit-based truncation of whole PKS systems, as well as targets for future study towards further boosting product titers.

## Results

**Design of engineering experiments based on classical modular boundaries.** The stambomycin PKS comprises 25 modules distributed among 9 polypeptides (Pks1–9)[33] (Fig. 1a) (Note: throughout the text, the stambomycin genes have been numbered in accordance with ref. [33]). Module 12 of Pks4 notably houses a broad-specificity AT domain which gives rise to the six characterized stambomycin family members (A–F), which differ in the alkyl functionality at the C-26 position[33,34]. To access abridged derivatives using the classical module boundaries, we reasoned that we could engineer intersubunit interfaces by suitable manipulation of docking domains. Encouragingly, the extreme C- and N-termini of all subunits (with the exception of the N-terminus of Pks1 and the C-terminus of Pks9) contain sequences with convincing homology to previously identified DDs[26,35] (the C-terminal DDs are referred to hereafter as $^CDDs$ and their partner N-terminal DDs as $^NDDs$). By bioinformatics analysis, we were able to confidently assign the DDs acting at 6 of the 8 interfaces to the type 1a class[26], and the remaining two sets of DDs as type 1b[35] (Supplementary Fig. 1). In both cases, docking occurs between an α-helical $^CDD$ and a coiled-coil formed by the $^NDD$, with specificity achieved via strategically placed charge:charge interactions at the complex interface (Supplementary Fig. 1)[26,35].

Among the type 1a junctions, there were notably two sets which appeared compatible in terms of the translocated substrate: Pks 3/4 + 7/8 and Pks 4/5 + 8/9 (Supplementary Fig. 2). Specifically, the functional groups at the critical α- and β-positions[17,36] of the transferred chains are identical at these junctions, and correspondingly, the downstream KSs show similarities across several sequence motifs previously correlated with substrate specificity[17,24,37] (Supplementary Fig. 2). Targeting such interfaces thus allowed us, at least in principle, to overcome the functional block to the engineered systems represented by poor recognition of the incoming substrate by the directly downstream KS domain[25]. Ultimately, we aimed to create an interface between Pks subunits 4 and 9 for two principal reasons. First, as mentioned earlier, Pks4 is at the origin of the structural variation between the stambomycin family members, and thus we anticipated that maintaining the subunit within the hybrid system would give rise to a corresponding series of truncated analogues, providing important evidence for their identities. Second, it was genetically more practical to modify the second set of interfaces

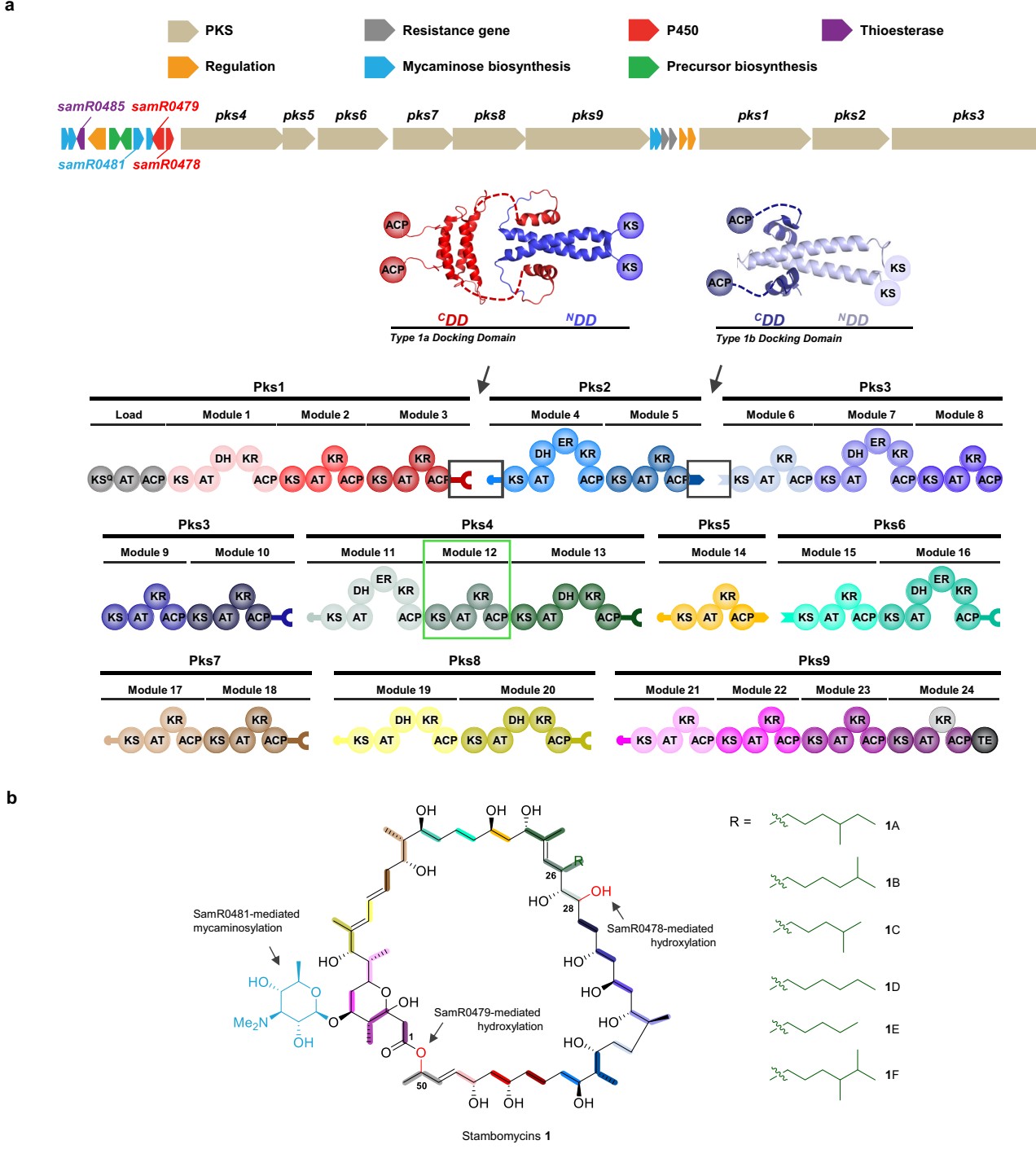

**Fig. 1 Stambomycin polyketide synthase and structures of stambomycin derivatives produced by *S. ambofaciens* ATCC23877. a** Organization of the stambomycin biosynthetic gene cluster, and schematic of the encoded polyketide synthase (PKS) subunits (Pks1–Pks9) showing the component modules and domains, as well as the intersubunit docking domains. The DDs belong to two distinct structural classes (type 1a and type 1b), for which representative NMR structures of complexes are shown[26,35]. The AT domain of PKS module 12 (green box) is responsible for recruiting six alternative extender units, resulting in a small family of stambomycins. The last KR domain of module 24 (gray) is inactive. **b** Structure of stambomycins **1** (A−F), which differ from each other in the alkyl functionality (R group) at position C-26 (the indicated stereochemistries[33] were predicted based on analysis of known domain stereochemical determinants[76], and those for the C-1–C-27 fragment recently confirmed by total synthesis[77]). The monomers are color-coded to match the modules responsible for their incorporation. The sites of glycosylation and hydroxylation are highlighted with their responsible enzymes indicated. KS ketosynthase (KS[Q] refers to replacement of the active site cysteine residue by glutamine), AT acyl transferase, ACP acyl carrier protein, DH dehydratase, ER enoyl reductase, KR ketoreductase, TE thioesterase, [C]DD C-terminal docking domain, [N]DD N-terminal docking domain.

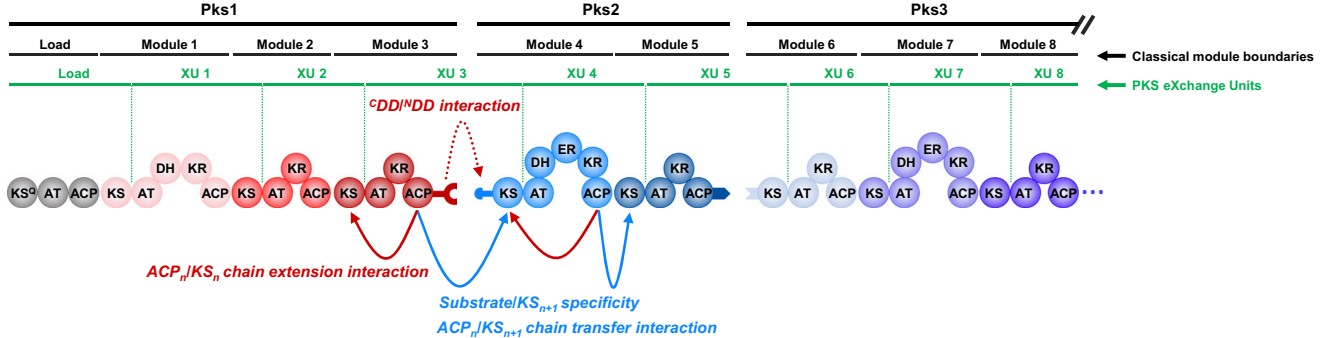

**Fig. 2 A schematic representation of classical and alternative module definitions.** A classical module (black) is defined as the catalytical unit responsible for incorporation of one building block into the growing polyketide chain, and associated functional group modifications. $^C$DD/$^N$DD pairs (shown) mediate communication between such traditionally defined modules. According to this definition, KS/ACP chain extension interactions (left-pointing arrows) occur within the modules, whereas ACP/KS chain transfer interactions (right-pointing arrows) occur between modules, and the incoming substrate for the KS domain is generated by the upstream module. PKS eXchange Units (XUs, shown in green) analogous to those used to engineer NRPS systems[19,65] were inspired by the evolutionary co-migration in certain systems of the KSs and the upstream processing domains[17,18]. Using these domains sets, the KS/ACP chain extension interaction (left-pointing arrows) is intermodular, while the ACP/KS chain transfer contacts (right-pointing arrows) are intramodular. KS ketosynthase (KS$^Q$ refers to replacement of the active site cysteine residue by glutamine), AT acyl transferase, ACP acyl carrier protein, DH dehydratase, ER enoyl reductase, KR ketoreductase, $^C$DD C-terminal docking domain, $^N$DD N-terminal docking domain.

due to splitting of the genes encoding the PKS subunits between two loci (Fig. 1 and Supplementary Fig. 4).

To establish the Pks4/Pks9 junction, we initially modified the $^C$DD of Pks4 ($^C$DD$_4$) to match that of Pks8 (the natural partner of the $^N$DD of Pks9 ($^N$DD$_9$)), either by site-directed mutagenesis (SDM) of residues previously identified as key mediators of interaction specificity (construct $^C$DD$_4$ SDM; Supplementary Fig. 3 and Supplementary Table 1)[26], or by exchange of the complete $^C$DD docking α-helix of $^C$DD$_4$ for that of $^C$DD$_8$ (construct $^C$DD$_4$ helix swap; Supplementary Fig. 3 and Supplementary Table 1)[38]. Modifying the $^C$DD$_4$ specificity code to match that of $^C$DD$_8$ required mutation of 3 residues, while for the $^C$DD$_4$ helix swap, the terminal 16 amino acids of $^C$DD$_4$ were exchanged for the corresponding 15 residues of $^C$DD$_8$ (Supplementary Fig. 3 and Supplementary Table 1). The genetic alterations were carried out in two distinct PKS contexts: (i) in the presence of the intervening subunits 5–8, which allowed for the possibility of competitive interactions between modified Pks4 and both Pks5 and Pks9; and (ii) removing the intervening subunits 5–8, thus eliminating competition for binding of Pks4 by Pks5, and of Pks9 by Pks8 (Supplementary Fig. 3). We further generated a mutant in which Pks subunits 5–8 were deleted but no modification was made to $^C$DD$_4$, in order to judge the intrinsic capacity of Pks4 and Pks9 to interact. Furthermore, genetic engineering was carried out in parallel by both PCR-targeting[39] and CRISPR-Cas9[40] (Supplementary Figs. 5 and 6), in order to directly compare the efficacy of these two approaches, as well as evaluate the effect of the short scar sequence remaining in the chromosome following PCR-targeting.

The $^C$DD$_4$ SDM and $^C$DD$_4$ helix swap sequences were introduced in parallel into the *S. ambofaciens* genome. As previous work has shown that production from the stambomycin biosynthetic gene cluster requires activation by constitutive overexpression of a pathway-specific LAL (Large ATP-binding regulators of the LuxR family) regulator[33], we additionally introduced the LAL over-expression plasmid (pOE484) into each of the mutants, using the empty parental plasmid (pIB139[41]) as a control. In total, this strategy resulted in 20 targeted strains harboring interface mutants (where K7N refers to PCR-targeting and CPN to CRISPR-Cas9 engineering): K7N1/pIB139, K7N1/OE484, K7N2/pIB139, K7N2/OE484, K7N3/pIB139, K7N3/OE484, K7N4/pIB139, K7N4/OE484, K7N5/pIB139, K7N5/OE484, K7N6/pIB139, K7N6/OE484, CPN1/pIB139, CPN1/OE484, CPN2/pIB139, CPN2/OE484, CPN4/pIB139, CPN4/

OE484, CPN5/pIB139, CPN5/OE484 (Table 1, Supplementary Data 1–3; despite extensive efforts the CPN3 mutant strain was not obtained). The principal difference between the K7N and CPN series of constructs is the presence of a 33 bp scar sequence between the modified *pks4* and *pks9* genes (Supplementary Fig. 4). Construct K7N6 was assembled specifically to test the effect of this region, without any further modification to $^C$DD$_4$ and the intervening *pks5*–*pks8* genes.

With the exception of K7N3, CPN4, and CPN5, extracts of the engineered mutant strains harboring pOE484 were analyzed by high performance liquid chromatography heated electrospray ionization high-resolution mass spectrometry (HPLC-ESI-HRMS) on a Dionex UItiMate 3000 HPLC coupled to a Q Exactive$^{TM}$ Hybrid Quadrupole-Orbitrap$^{TM}$ Mass Spectrometer, and compared to extracts of the control strain containing pIB139[41] as well as the wild-type *S. ambofaciens*, using SIEVE 2.0 screening software. All extracts were subsequently analyzed on a Thermo Scientific Orbitrap LTQXL and/or an Orbitrap ID-X Tribrid Mass Spectrometer, and the data inspected manually. Metabolites not present in the control strains are listed in Table 1 and Supplementary Table 2.

It must be noted that the low yields of the target mini-stambomycins (vide infra) precluded their purification, and consequently full structure elucidation by NMR and use as quantification standards. However, we were able to obtain convincing evidence for their identities based on exact masses obtained from high-resolution mass spectra, and detailed comparative analysis of MS and MS$^2$ data with those acquired on a series of shorter derivatives which were produced in substantially higher amounts (as detailed in the respective Supplementary figures). Further support for the identity of multiple metabolites was afforded by genetic engineering controls. In the absence of authentic standards for the stambomycin derivatives, we evaluated the use of two surrogates for quantification: the parental stambomycins **1**A/B[33] (Supplementary Fig. 7–9), and linear, 50-deoxy derivatives of stambomycins **2**A/B (Supplementary Figs. 10 and 11). The latter compounds were purified from a previously described strain in which the C-50 hydroxylase SamR0479 (Fig. 1a) had been inactivated[42]. This analysis notably revealed that the detection sensitivity towards the 50-deoxystambomycins **2** whether using MS or UV, was dramatically lower than for the parental stambomycins **1** (Supplementary Fig. 11). In the case of the MS analysis, we can clearly attribute this difference to the presence in **1** of β-D-mycaminose which contains an easily protonatable

**Table 1 Summary of various strains generated, as well as the stambomycin derivatives detected.**

| Strain | Modifications introduced | Stambomycins 1 | Derivatives detected |
|---|---|---|---|
| ATCC/OE484 | Wild-type | ✓ | n.d. |
| K7N6/OE484[a] | 33 bp scar[c] | ✓ | n.d. |
| K7N5/OE484 | $^{C}DD_4$ helix swap, 33 bp scar | ✗ | **4, 5, 6, 7** |
| CPN5/OE484[b] | $^{C}DD_4$ helix swap | ✓ | n.d. |
| K7N4/OE484 | $^{C}DD_4$ site-directed mutagenesis (SDM), 33 bp scar | ✓ | n.d. |
| CPN4/OE484 | $^{C}DD_4$ SDM | ✓ | n.d. |
| K7N3/OE484 | $\Delta pks5–8$, 33 bp scar | ✗ | **4, 5, 6, 7** |
| K7N2/OE484 | $^{C}DD_4$ helix swap, $\Delta pks5–8$, 33 bp scar | ✗ | **4, 5, 6, 7** |
| CPN2/OE484 | $^{C}DD_4$ helix swap, $\Delta pks5–8$ | ✗ | **4, 5, 6, 7** |
| K7N1/OE484 | $^{C}DD_4$ SDM, $\Delta pks5–8$, 33 bp scar | ✗ | **4, 5, 6, 7** |
| CPN1/OE484 | $^{C}DD_4$ SDM, $\Delta pks5–8$ | ✗ | **4, 5, 6, 7** |
| ATCC/OE484/Pks4+TEI | TEI fused to Pks4 | ✗ | **4, 5, 6, 7** |
| CPN2/OE484/TEI SDM | TEI inactivation (Ser → Ala), $^{C}DD_4$ helix swap, $\Delta pks5–8$ | ✗ | **4, 5, 6, 7** |
| CPN2/OE484/TEII SDM | TEII inactivation (Ser → Ala), $^{C}DD_4$ helix swap, $\Delta pks5–8$ | ✗ | **4, 5, 6, 7** |
| CPN2/OE484/Δ478 | $\Delta samR0478$, $^{C}DD_4$ helix swap, $\Delta pks5–8$ | ✗ | **4, 5, 6, 7** |
| CPN2/OE484/Δ479 | $\Delta samR0479$, $^{C}DD_4$ helix swap, $\Delta pks5–8$ | ✗ | **8, 9, 10, 11** |
| CPN2/OE484/ACP$_{13}$ SDM | $ACP_{13}$ H1 modified[d], $^{C}DD_4$ helix swap, $\Delta pks5–8$ | ✗ | **4, 5, 6, 7, 13** |
| ATCC/OE484/hy59_S1 | $^{N}DD_9 + KS_{21}$ replaced by $^{N}DD_5 + KS_{14}$, $\Delta pks5–8$ | ✗ | **4, 5, 6, 7, 12, 13, 14** |
| ATCC/OE484/hy59_S2 | $^{N}DD_9 + KS_{21}$ replaced by $^{N}DD_5 + KS_{14/21}$, $\Delta pks5–8$ | ✗ | **4, 5, 6, 7, 12, 13, 14, 16** |
| ATCC/OE484/hy59_S1/Δ479 | $\Delta samR0479$, $^{N}DD_9 + KS_{21}$ replaced by $^{N}DD_5 + KS_{14}$, $\Delta pks5–8$ | ✗ | **8, 9, 10, 11, 15** |
| ATCC/OE484/hy59_S2/Δ479 | $\Delta samR0479$, $^{N}DD_9 + KS_{21}$ replaced by $^{N}DD_5 + KS_{14/21}$, $\Delta pks5–8$ | ✗ | **8, 9, 10, 11, 15, 17** |
| ATCC/OE484/hy59_S1/ACP$_{21}$ region swap | $ACP_{21}$ L1 + H2 modified[e], $^{N}DD_9 + KS_{21}$ replaced by $^{N}DD_5 + KS_{14}$, $\Delta pks5–8$ | ✗ | **4, 5, 6, 7** |
| ATCC/OE484/hy59_S1/ACP$_{21}$ GtoD | $ACP_{21}$ L1 modified[f], $^{N}DD_9 + KS_{21}$ replaced by $^{N}DD_5 + KS_{14}$, $\Delta pks5–8$ | ✗ | **4, 5, 6, 7, 13, 16** |

n.d. indicates no stambomycin derivatives were detected.
[a]K7N (pronounced 'cassette number') refers to mutants generated by PCR-targeting.
[b]CPN refers to mutants generated using CRISPR-Cas9.
[c]Use of PCR-targeting technique introduced a 'scar' sequence between *pks4* and *pks9*; for details, see Supplementary Fig. 4.
[d]H1 modified refers to mutation of six residues within the helix α1 region of $ACP_{13}$ (EADQRR to PSERRQ); for details, see Supplementary Fig. 29.
[e]L1 + H2 modified refers to exchange of the loop 1 + helix α2 region of $ACP_{21}$; for details, see Supplementary Fig. 39.
[f]L1 modified indicates that one residue within the loop 1 region of $ACP_{21}$ was mutated ($G_{1499}$ to D); for details, see Supplementary Fig. 39.

nitrogen, as it is absent in **2**. Indeed, analysis of erythromycin A **3** which contains an alternative amino sugar, β-D-desosamine, showed it to be detected with similar sensitivity to **1** (Supplementary Fig. 12). Thus, overall, to permit an estimation of yield ranges for the engineered metabolites, we generated a standard curve based on stambomycins **1**A/B for which we could detect a 25,000-fold range of concentrations (0.00001–0.25 mg mL$^{-1}$). Using this curve directly then provided the lower yield limit for the derivatives, while introduction of a correction factor (×206) based on the 50-deoxystambomycins **2**, furnished the upper yield limit. Importantly, the maximum yields calculated directly from a limited calibration curve based on **2**, did not differ substantially from those determined using the correction factor (Supplementary Table 3).

The first result is that the K7N6/OE484 mutant yielded a similar metabolic profile to *S. ambofaciens* wt ($22 \pm 3$ mg L$^{-1}$ of stambomycins **1** (Supplementary Table 4), 73% relative yield), showing that the scar sequence impacted stambomycin production, but not dramatically (Fig. 3). By contrast, no stambomycins were observed, as anticipated, in all constructs in which Pks5–Pks8 had been removed (K7N1−3; CPN1, 2) (Fig. 3). Stambomycins **1** were present, however, in strains K7N4 and CPN4 harboring $^{C}DD_4$ site-directed mutations and in the $^{C}DD_4$ helix swap strain CPN5, all of which still contained Pks5–Pks8, albeit at reduced amounts relative to the wild-type (18, 23, and 14% of wt, respectively) (Fig. 3 and Supplementary Table 4). (Surprisingly, the metabolic profile of K7N5 reproducibly differed from that of CPN5, as no stambomycin-related metabolites were detected from K7N5 (Fig. 3)). These data suggested that while the mutations introduced into $^{C}DD_4$ negatively impacted the interaction with $^{N}DD_5$, they were not sufficient to disrupt natural chain transfer between Pks4 and Pks5. Thus, DD engineering to

alter partner choice should be accompanied by removal of competing intersubunit interactions.

We did not find any evidence in the DD engineering experiments for any of the target 37-membered metabolites (Supplementary Figs. 3 and 13). However, all strains in which stambomycin production was abolished (Table 1) exhibited four peaks in common (Fig. 3b and Supplementary Fig. 13) (peaks potentially corresponding to additional derivatives were observed, but none were shared between multiple strains). The determined exact masses and MS/MS analysis (as exemplified by strain CPN2/OE484, Fig. 3b) correspond to truncated derivatives of stambomycins A/B and C/D respectively, following premature release from modules 12 and 13 of Pks4 (compounds **4–7**, Fig. 3d and Supplementary Figs. 13–15; ca. 8-fold greater yield of the module 13 products (Supplementary Table 5)). Further support for the identity of these shunt compounds was obtained by grafting the chain-terminating (type I) thioesterase (TE) domain from the C-terminal end of Pks9 to the C-terminus of Pks4 in order to force chain release at this stage. Indeed, identical compounds were produced, but at 17-fold increased yield relative to CPN2/OE484, consistent with active off-loading of the chains (Fig. 3c, Supplementary Figs. 16–18, and Supplementary Table 5).

Based on their exact masses, both sets of shunt metabolites were hydroxylated on a single carbon, while none were found to bear the β-mycaminose of the mature stambomycins, consistent with the absence of the tetrahydropyran moiety to which it is normally tethered. To determine the location of the hydroxylation and therefore the hydroxylase responsible, we inactivated in mutant CPN2/OE484 the genes *samR0478* and *samR0479* encoding respectively, the stambomycin C-28 and C-50 cytochrome P450 hydroxylases (Fig. 1a)[42]. While extracts of CPN2/OE484/Δ478 were

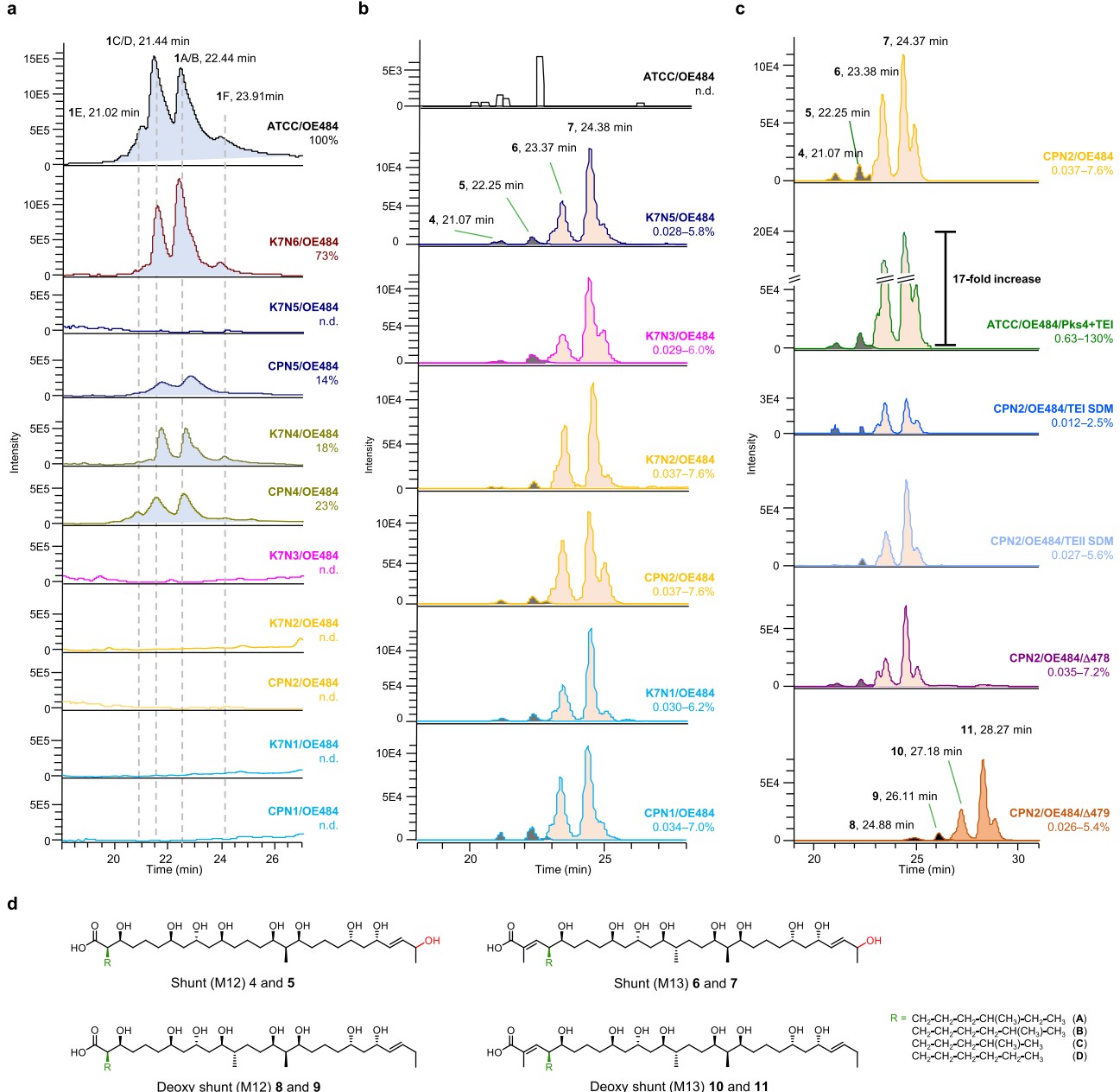

**Fig. 3 Analysis of metabolites derived from PKS engineering based on the classical module definition. a** HPLC-PDA analysis at $\lambda_{max}$ 238 nm of stambomycins **1** from the wild-type strain and various mutants. **b** LC-ESI-HRMS analysis of mutants in which **1** was absent revealed a series of shunt products (**4**–**7**). Shown are the extracted ion chromatograms (EICs) of **4**–**7**, using the calculated $m/z$ shown in Supplementary Tables 4 and 5. The indicated estimated yield ranges are derived from the stambomycins **1**A/B calibration curve (lower limit) and the 50-deoxystambomycins **2**A/B correction factor (upper limit) (Supplementary Figs. 9 and 11 and Supplementary Tables 4 and 5). **c** LC-ESI-HRMS analysis of several CPN2-derived mutants (the yields of shunt products **4**–**7** are shown relative to **1** in the wt (average of four measurements)). Notably, the combined yield of **4**–**7** in mutant ATCC/OE484/Pks4+TEI was 17-fold higher than that from CPN2/OE484. A series of compounds **8**–**11** was generated in strain CPN2/OE484 in which the gene samR0479 was deleted. **d** Chemical structures of shunts **4**–**11**. The structural differences among the metabolites are highlighted (green = R group; red = hydroxyl). Shunt products **4**, **6**, **8**, and **10** correspond to stambomycin C/D derivatives, and **5**, **7**, **9**, and **11** to stambomycin A/B derivatives. M12 and M13 indicate shunt compounds released from modules 12 and 13, respectively. As in Fig. 1, the indicated configurations have been extrapolated from those assigned to the stambomycins **1**[77]. TE thioesterase, SDM site-directed mutagenesis.

unchanged relative to CPN2/OE484 (i.e., the hydroxyl group was still present), the CPN2/OE484/Δ479 mutant exhibited four peaks with masses and fragmentation patterns corresponding to the deoxy shunt products (Fig. 3, Supplementary Figs. 19–21 (compounds **8**–**11**) and Supplementary Table 5). Taken together, these data show that the unusual online modification catalyzed by SamR0479[42], which is

necessary for macrocyclization, occurs prior to chain extension by Pks5. While SamR0478 has also been speculated to act during chain assembly[42], hydroxylation evidently occurs downstream of Pks4, at least. The intriguing substrate structural and/or protein–protein recognition features controlling the timing of hydroxylation by these P450 enzymes remain to be elucidated.

**Role of TE domains in release of the shunt metabolites**. We attributed the observed shunt metabolites to the lack of productive chain translocation between Pks4 and Pks9, causing intermediates to accumulate on ACPs 12 and 13. To evaluate whether these were released by spontaneous hydrolysis or enzymatically, we further investigated the role of the Pks9 TEI[42] in chain release, as well as that of SamR0485, a proof-reading type II TE[43] located in the cluster. Both TEs were disabled by site-directed mutagenesis of the active site serines (Ser to Ala) (Supplementary Fig. 17).

Interestingly, inactivation of both the type I and type II TEs reduced the yields of shunt products **4–7** relative to the parental strain CPN2/OE484 (by 66% and 27%, respectively; average of duplicate experiments) (Supplementary Fig. 18 and Supplementary Table 5). These data clearly show that premature release of the chains is catalyzed, at least in part, by both TEs in the cluster, although spontaneous liberation also occurs. While type II TEs typically interact with acyl-ACPs in trans to release blocked chains[43], the effect of the Pks9 TEI is less readily explained. One possibility is that the productive docking interaction between Pks4 and Pks9 allows Pks9 to adopt an alternative conformation from which the TE can off-load intermediates bound to Pks4 $ACP_{12}$ and $ACP_{13}$ (Supplementary Fig. 18).

Although this mechanism is reminiscent of that used by the pikromycin PKS to generate both 12- and 14-membered rings[44], the pikromycin TEI is separated from its alternative ACP target by a single module, while Pks9 TEI is located five or four modules downstream from ACPs 12 and 13 in the engineered system, which would seem to necessitate substantial intersubunit acrobatics. Alternatively, or in addition, such remote off-loading may involve interactions between distinct assembly lines within the context of a PKS megacomplex, as described for the bacillaene system of *Bacillus subtilis*[45].

**Understanding the docking domain engineering via studies in vitro with recombinant domains**. To better understand the results of the DD engineering, we studied in vitro the wild-type DD pairs ($^{C}DD_4/^{N}DD_5$ and $^{C}DD_8/^{N}DD_9$), as well as binding between the modified versions of $^{C}DD_4$ and wild-type $^{N}DD_9$. Design of suitable expression constructs in *E. coli* (Supplementary Table 1 and Supplementary Data 1 and 2) was based on bioinformatics analysis of the C-terminal ends of Pks4 and Pks8, and the N-termini of Pks5 and Pks9, and secondary structure analysis using PSIPRED[46] (Supplementary Fig. 22). Overall, we expressed and purified the following proteins in recombinant form from *E. coli*: $^{C}DD_4$ wt, $^{C}DD_4$ SDM, $^{C}DD_4$ helix swap, $^{N}DD_5$, and $^{C}DD_8$ (Supplementary Figs. 22 and 23, Supplementary Data 3). As $^{N}DD_9$ proved insoluble when expressed in *E. coli*, two versions with alternative start sites were obtained as synthetic peptides (Met and Val; Supplementary Fig. 22, Supplementary Table 1). Analysis of the individual $^{C}DDs$ by circular dichroism (CD) confirmed their expected high α-helical content ($^{C}DD_4$ wt (100 μM): 58%; $^{C}DD_8$ wt (100 μM): 49%), and showed no evident effect of the introduced mutations on secondary structure (Supplementary Fig. 24). All of the constructs were further confirmed to be homodimeric by size exclusion chromatography multi-angle light scattering (SEC-MALS) (Supplementary Fig. 25).

The two $^{N}DDs$ also exhibited α-helical character, though less pronounced than the $^{C}DDs$ ($^{N}DD_5$ (100 μM): 27%; $^{N}DD_9$ Met (100 μM): 21%; $^{N}DD_9$ Val (100 μM): 25%), and were monomeric by SEC-MALS (Supplementary Fig. 25). The latter result was surprising as type 1a $^{N}DDs$ classically form a homodimeric coiled-coil domain (Fig. 1, Supplementary Fig. 1), but we recently identified functional, monomeric type 1 $^{N}DDs$[47]. Indeed, we detected binding between the native pairs by isothermal titration calorimetry (ITC), with

affinities in the range of those determined previously for matched pairs of DDs[35,47–49] ($^{C}DD_4 + ^{N}DD_5$, $K_d = 14.5 ± 0.9$ μM; $^{C}DD_8 + ^{N}DD_9$ Met, $K_d = 33 ± 2$ μM; $^{C}DD_8 + ^{N}DD_9$ Val, $K_d = 22 ± 1$ μM) (Supplementary Fig. 26). Thus, while stable homodimerization of the $^{N}DDs$ may depend on the presence of a downstream homodimeric KS domain, their monomeric character did not preclude interaction with their $^{C}DD$ partners. Based on the higher affinity of the interaction, we could identify the $^{N}DD_9$ Val as the physiologically relevant construct. The observed binding stoichiometry (1 homodimeric $^{C}DD$:2 monomeric $^{N}DDs$), is consistent with the known structure of a type 1a complex in which two monomers of each DD are present (Fig. 1, Supplementary Fig. 1)[26]. As expected, no nonspecific interaction was detected between native $^{C}DD_4$ and $^{N}DD_9$, explaining the lack of productive communication between subunits Pks4 and Pks9 when the intervening multi-enzymes are deleted (strain K7N3) (Fig. 3a).

Analysis by ITC of binding between $^{C}DD_4$ SDM or $^{C}DD_4$ helix swap and $^{N}DD_5$ revealed the complete absence of interaction (Supplementary Fig. 26), and therefore that the introduced modifications were sufficient to disrupt communication between the native pair. Thus, the continued production of stambomycins **1** by K7N4, CPN4, and K7N5 harboring Pks5–Pks8 must be due to additional contacts between Pks4 and Pks5 beyond the docking domains, likely including the compatible $ACP_{13}/KS_{14}$ interface[15]. On the other hand, no interaction was detected between $^{C}DD_4$ SDM and $^{N}DD_9$, showing that this limited number of mutations was inadequate to induce productive contacts. This result is fully in accord with the absence of the expected mini-stambomycin products from these strains (K7N1/CPN1, Fig. 3a). By contrast, the $^{C}DD_4$ helix swap exhibited essentially the same binding to $^{N}DD_9$ Val as $^{C}DD_8$ ($K_d = 21.0 ± 0.3$ μM), demonstrating that exchange of just this helix is sufficient to redirect docking specificity[38]. Thus, inefficient docking is not at the origin of the failure of the $^{C}DD_4$ helix swaps to yield chain-extended products in vivo (strains K7N2/CPN2, Fig. 3a). We could therefore conclude at this stage that the problem arose from the non-native interface generated between $ACP_{13}$ and $KS_{21}$, poor acceptance by $KS_{21}$ of the incoming substrate during chain transfer and/or chain extension, and/or low activity towards the modified intermediate of domains and modules acting downstream.

**Attempted optimization of the stambomycin DD mutants**. We aimed next to improve the engineered Pks4/Pks9 intersubunit interface in strain CPN2 ($^{C}DD_4$ helix swap + deletion of Pks5–8) by targeting helix αI of $ACP_{13}$, as the first 10 residues of this helix have been implicated previously in governing the interaction with the downstream KS domain at hybrid junctions[50]. Notably, multiple sequence alignment of all ACPs in the stambomycin PKS located at intersubunit junctions revealed a unique sequence for each ACP in the helix αI region. This observation is consistent with a recognition code for the KS partner, and the idea that mismatching these contacts might hamper productive chain transfer (Supplementary Fig. 27). Indeed, as mentioned previously, even when docking is interrupted, contacts between $ACP_{13}$ and $KS_{14}$ are apparently sufficient to enable chain translocation between Pks4 and Pks5 (Fig. 3a). In addition, an analogous strategy of optimizing the $ACP_n/KS_{n+1}$ chain transfer interface was shown recently to substantially improve interaction between an ACP (JamC) derived from the jamaicamide B biosynthetic pathway, and the first chain extension module of the lipomycin PKS (LipPKS1)[51].

In our case, the first six residues of $ACP_{13}$ helix αI were modified using CRISPR-Cas9, so that the full 10-residue recognition sequence matched that of $ACP_{20}$, the natural partner of $KS_{21}$ (EADQRR → PSERRQ) (Supplementary Figs. 27 and 28).

Analysis of extracts of the resulting strain CPN2/OE484/ACP$_{13}$ SDM by LC-ESI-HRMS revealed at best small amounts (maximum yield of 0.1 mg L$^{-1}$) of target cyclic mini-stambomycins A/B (**13**), lacking the hydroxyl group introduced by SamR0478 (Fig. 4, Supplementary Fig. 29 and Supplementary Table 6). Thus, while this experiment finally yielded evidence for successful chain transfer between Pks4 and Pks9 followed by subsequent chain extension by Pks9 and TE-catalyzed release, the overall efficiency of the system remained low. Interestingly, however, the titers of the four shunt metabolites **4–7** were as much as 48-fold higher from the ACP$_{13}$ helix swap mutant than from CPN2/OE484. Evidently, improved interactions between ACP$_{13}$ and KS$_{20}$ facilitated release of the stalled intermediates from ACPs 12 and 13, presumably via remote action by the TEI domain.

**Engineering mini-stambomycins by maintaining the native ACP$_{13}$/KS$_{14}$ junction.** Cumulatively, the results obtained with the docking domain engineering identified KS$_{21}$ as one potential bottleneck in the engineered PKS. Our parallel strategy based on PKS XUs (Fig. 2) allowed us to directly test this idea. Specifically, we investigated the effects of preserving the native $^{C}$DD$_4$/$^{N}$DD$_5$ pair and either the majority of KS$_{14}$, or a little more than half of the domain, resulting in a KS$_{14}$/KS$_{21}$ hybrid. For this, we used two different splice sites in KS$_{14}$ (Supplementary Fig. 30): (i) at the end of the domain in a highly conserved region (GTNAHV) exploited recently to efficiently swap downstream AT domains[52]; and, (ii) at a site corresponding to a recombination hot spot identified during induced evolution of the rapamycin (RAPS) PKS[53], yielding the KS$_{14}$/KS$_{21}$ chimera (Fig. 4 and Supplementary Fig. 31). Both of these modifications were introduced into *S. ambofaciens* using CRISPR-Cas9, while simultaneously removing Pks5–Pks8, giving, respectively, after co-transformation with pOE484 and the control plasmid pIB139, strains ATCC/OE484/hy59_S1, ATCC/pIB139/hy59_S1, ATCC/OE484/hy59_S2, and ATCC/pIB139/hy59_S2.

Analysis of culture extracts relative to the controls revealed the presence in both ATCC/OE484/hy59_S1 and ATCC/OE484/hy59_S2, of a series of 37-membered metabolites (Fig. 4). The obtained comprehensive MS/MS data were consistent with the desired mini-stambomycins either as their free acids or in cyclic form (metabolites **12–14**, Fig. 4, Supplementary Figs. 32–35 and Supplementary Note 1). Signals corresponding to the A/B and C/D derivatives of all metabolites were detected, providing important evidence for their identities, as well as both the C-14 hydroxylated **14** and non-hydroxylated **13** forms of the cyclic mini-stambomycins (C-14 corresponds to C-28 in the parental compounds (Fig. 1)). For detailed justification of the structure assignments of **13** and **14**, see the Supplementary Note 1. It is not surprising that the corresponding E and F forms were not detected, as their yields even from the wild-type are much lower than the A–D derivatives (Fig. 3a). Critically, we obtained additional support for the identities of **12–14** by inactivation of *samR0479* (which introduces the hydroxyl used for macrocyclization), which resulted in exclusive production of linear deoxy mini-stambomycins **15** (Supplementary Figs. 36–38 and Supplementary Table 7). The observation of non-hydroxylated **13** shows notably that internal hydroxylation by SamR0478 is not an absolute prerequisite for TE-catalysed macrolactonization, and argues that hydroxylation of the mini-stambomycins only takes place on the macrocyclic compound. Although compounds **13** and **14** likely incorporate the tetrahydropyran moiety of the parental stambomycins **1**, which undergoes glycosylation, derivatives bearing β-D-mycaminose were not observed, presumably due to poor recognition of the overall modified macrocycle by glycosyl transferase SamR0481[33].

The combined, estimated maximum yields of the target compounds were reduced relative to the wild-type stambomycins by some 8-fold, and variable between fermentations. Notably,

however, metabolites **12–14** were obtained at approximately three-fold higher titer from ATCC/OE484/hy59_S2 incorporating the hybrid KS$_{14}$/KS$_{21}$ (0.76, 2.0, and 0.74 mg L$^{-1}$, respectively, (3.5 mg L$^{-1}$ total)) than from ATCC/OE484/hy59_S1 containing the full KS$_{14}$ swap (0.27, 0.66, and 0.20 mg L$^{-1}$, respectively (1.1 mg L$^{-1}$ total)) (Fig. 4, Supplementary Fig. 32 and Supplementary Table 6). As observed previously, the strains also produced substantial quantities of the shunt products **4–7**, while inactivation of *samR0479* led correspondingly to the deoxy versions of these compounds **8–11** (Supplementary Figs. 32 and 36). The yields of the shunts were ca. 80-fold higher than those of the corresponding mini-stambomycins, with the highest titers observed in the strain incorporating the hybrid KS$_{14}$/KS$_{21}$. The amount of shunt metabolites was also ~123-fold higher than from strain CPN2/OE484 (which incorporates an ACP$_{13}$-$^{C}$DD$_4$ swap/$^{N}$DD$_9$-KS$_{21}$ interface) (Figs. 3b and 4, Supplementary Table 6). Thus, contrary to expectation, although using the KS as a fusion site improved communication between Pks4 and Pks9, it also substantially boosted TEI-mediated off-loading of stalled upstream intermediates.

In principle, such stalling could result from a slow rate of chain extension in the now hybrid acceptor module (for example, in the full KS swap construct, KS$_{14}$ and ACP$_{21}$ are completely mismatched for chain extension). To evaluate this idea, we modified ACP$_{21}$ within ATCC/OE484/hy59_S1 incorporating the full-length KS$_{14}$, targeting a sequence region previously identified as mediating intramodular communication between the KS and ACP during chain extension (Supplementary Fig. 39)[23,50]. Specifically, we exchanged loop 1 and the initial portion of helix αII of ACP$_{21}$ for the corresponding sequence of ACP$_{14}$, using CRISPR-Cas9 (Supplementary Fig. 39). As we anticipated that creation of this substantially hybrid ACP might engender structural perturbation, we also engineered a minimal mutant of ACP$_{21}$ in which only one of the two most critical residues in the recognition motif was mutated to the corresponding amino acid in ACP$_{14}$ (G$_{1499}$ of Pks9 → D; the second residue, R, of the motif is already common to the two ACPs) (Supplementary Fig. 39). Analysis of the loop/helix αII swap by HPLC-MS showed that all mini-stambomycin production had been abolished (Supplementary Fig. 40 and Supplementary Table 6), consistent with the anticipated disruption to ACP$_{14}$ structure. Production by the ACP site-directed mutant was not any better than by the full KS swap construct (Fig. 4, Supplementary Fig. 40 and Supplementary Table 6), as only metabolite **13** remained detectable.

In principle, the hybrid KS$_{14}$/KS$_{21}$ domain may have worked better than KS$_{14}$ for chain extension due to improved interaction with ACP$_{21}$, with stalling displaced to later modules. If this were the case, we might expect to see accumulation in the medium of shunt metabolites corresponding to the intermediate generated by module 21. Indeed, in the case of strain hy59_S2 (chimeric KS$_{14}$/KS$_{21}$) but not hy59_S1 (KS$_{14}$), we detected masses consistent with the A/B and C/D forms of intermediate **16** generated by module 21, at yields comparable to those of the final mini-stambomycins (Fig. 4, Supplementary Fig. 41 and Supplementary Table 6). Correspondingly, **17**, the C30-deoxy analogue of **16**, was detected in the SamR0479 mutant (Supplementary Figs. 37 and 38 and Supplementary Table 7). The same metabolite **16** was identified from the ACP$_{21}$ G →D mutant (Fig. 4 and Supplementary Fig. 40 and Supplementary Table 6), consistent with interrupted chain transfer to KS$_{22}$. Taken together, these data confirm module 22 as a blockage point in the engineered systems.

**Relative efficacy of PKS engineering using PCR-targeting and CRISPR-Cas9.** As multiple of our core constructs were generated by both PCR-targeting and CRISPR-Cas9, we were able to

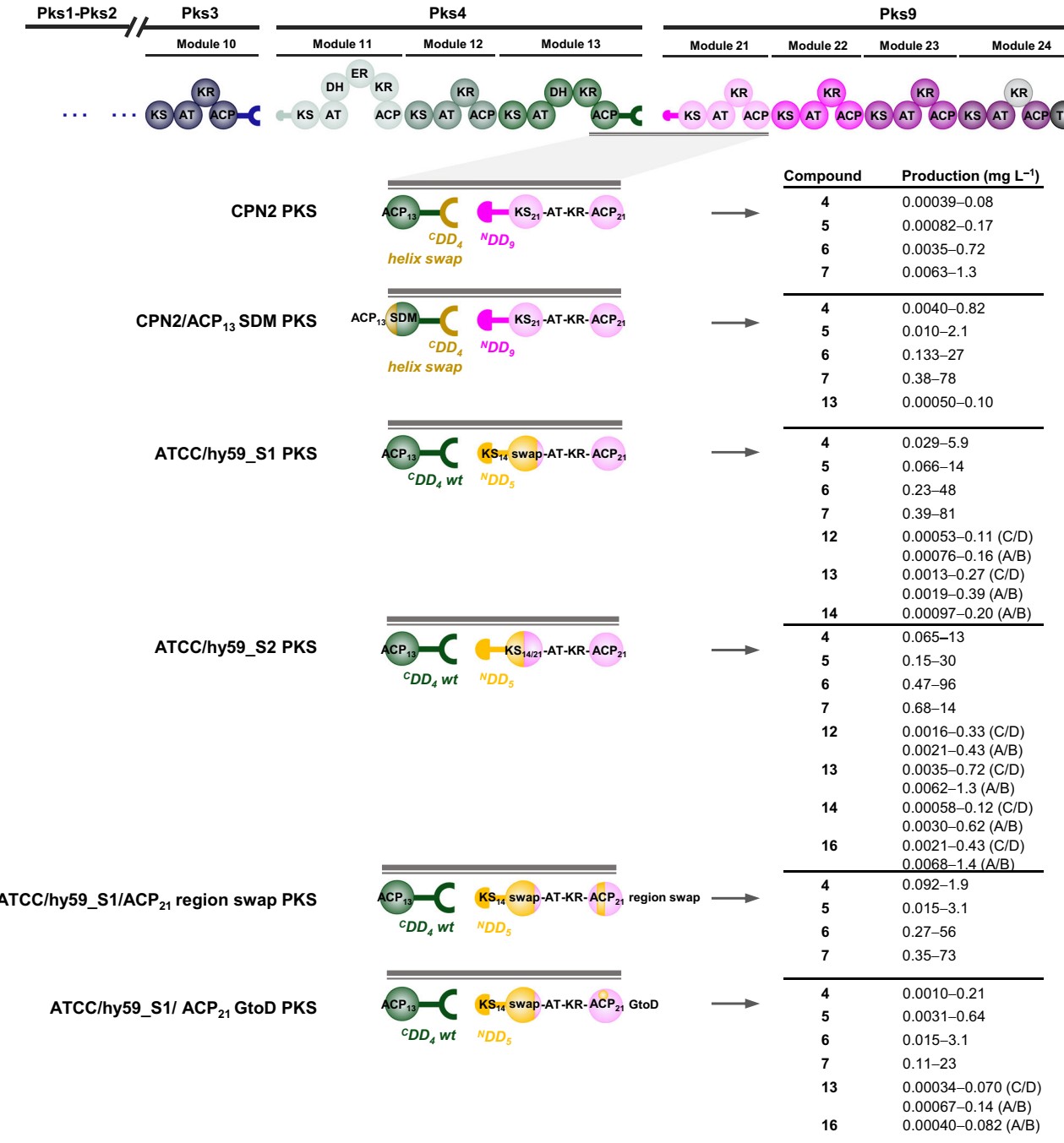

**Fig. 4 Engineering of functional mini-stambomycin PKSs.** The various strategies used in each case are represented schematically, along with the obtained products and their estimated yield ranges (full analysis of all constructs is provided in Supplementary Table 6). The engineering starting point, CPN2 PKS, contains a functional $^{C}DD_4$ helix swap/$^{N}DD_9$ docking interaction (swapped docking α-helix shown in dark yellow), but a mismatched $ACP_{13}$/$KS_{21}$ interdomain interaction. This PKS yielded only shunt products **4–7**. The CPN2/$ACP_{13}$ SDM PKS, in which the $ACP_{13}$ helix αI has been modified to match that of $ACP_{20}$ (dark yellow ball), generates mini-stambomycin derivatives (**13**, a cyclic form lacking the internal hydroxy, Supplementary Fig. 29). The ATCC/hy59_S1 and S2 constructs were based on the PKS exchange unit (XU) concept, as the engineering point was selected downstream of the $^{C}DD_4$/$^{N}DD_5$ interface within the $KS_{21}$ domain. Of the two junctions, that in which the fusion was located essentially at the mid-point of the domain (ATCC/hy59_S2) functioned better than that which included the majority of $KS_{14}$ (ATCC/hy59_S1), although both PKSs led to successful generation of three mini-stambomycins, both as their free acids (**12**) and in macrolide form (**13** and **14**) (Supplementary Fig. 32). In an attempt to boost yields from ATCC/hy59_S1, two further constructs were created by modification of ACP21-either by swapping a region implicated in KS/ACP communication during extension (ATCC/hy59_S1/$ACP_{21}$ region swap), or by mutating a single key residue within this motif (ATCC/hy59_S1/$ACP_{21}$ GtoD). The ATCC/hy59_S1/$ACP_{21}$ region swap yielded only the shunt metabolites **4–7**, while solely **13** among **12-14** was detected from ATCC/hy59_S1/$ACP_{21}$ GtoD, showing that the introduced changes did not work as intended (Supplementary Fig. 39). We observed in addition from ATCC/hy59_S2 PKS and ATCC/hy59_S1/$ACP_{21}$ GtoD, shunt product **16** corresponding to the chain released from module 21 (Supplementary Figs. 40 and 41), identifying the downstream module 22 as a blockage point. KS ketosynthase ($KS^{Q}$ refers to replacement of the active site cysteine residue by glutamine), AT acyl transferase, ACP acyl carrier protein, DH dehydratase, ER enoyl reductase, KR ketoreductase, TE thioesterase, $^{C}DD$ C-terminal docking domain, $^{N}DD$ N-terminal docking domain.

directly compare the efficiency of the two techniques (Fig. 3 and Supplementary Figs. 5 and 6). Globally, our results confirm that both approaches can be employed to introduce large-scale modifications to PKS biosynthetic genes (i.e., deletions of single or multi-gene regions)[40,54–56]. We have also demonstrated that CRISPR-Cas9 can be leveraged to specifically modify modular PKS domains[57]. Of the two methods, CRISPR-Cas9 was the more rapid, as the corresponding constructs were engineered in approximately half of the time. In addition, while CRISPR-Cas9 allowed for direct modification of the host genome, PCR-targeting relied on the availability of suitable cosmids housing the target genes, and resulted in a 33 bp *attB*-like scar sequence in the genome (Supplementary Fig. 4)[58]. In addition to hampering iterative use of the approach, the scar apparently provoked a moderate reduction in stambomycin yields in mutant K7N6 compared to the wild-type, an effect also noted upon comparison of several analogous mutant strains (e.g., K7N4 vs. CPN4, Fig. 3). Nonetheless, we did encounter certain difficulties with use of CRISPR-Cas9 (i.e., failure to obtain construct CPN3, occasional reversions to wild-type, etc.), observations motivating ongoing efforts in other laboratories to further enhance the suitability of CRISPR-Cas9 for editing PKS pathways[57,59–64].

## Discussion

In this work, we have utilized an approach based on the state-of-the-art in PKS engineering to modify the stambomycin PKS (Fig. 5). Specifically, we aimed to remove the four PKS subunits between Pks4 and Pks9 in the assembly line which together house seven chain extension modules, to generate a series of 37-membered mini-stambomycins. While in principle such a change might have been possible by directly fusing Pks4 and Pks9 via a suitable intermodular linker, this approach would have resulted in a heptamodular subunit whose size is far in excess of the tetra-modular multienzymes present in the system. We have also demonstrated recently the low efficacy of this strategy when the module downstream of the linker is N-terminal in its native subunit context, as with module 21 of Pks9[22].

As an initial approach (Fig. 5), we modified $^{C}DD_4$ to render it compatible with $^{N}DD_9$. The aim in this case was to induce productive communication between Pks4 and Pks9, while leaving all modular units intact. This modified PKS relied for function on both a non-native chain transfer interface ($ACP_{13}/KS_{21}$), and the intrinsic tolerance of the downstream KS/modules to the incoming substrate. We were optimistic this experiment might work given the structural similarities between the native substrates of $KS_{14}$ and $KS_{21}$ at least directly adjacent to the acyl terminus, as well as the fact that the stambomycin PKS generates a small family of metabolites, and therefore must exhibit some intrinsic tolerance to structural variation. Although we showed in vitro with recombinant DD pairs that a docking helix-swapped mutant of $^{C}DD_4$ communicated effectively with $^{N}DD_9$, chain transfer across the engineered interface did not occur in vivo, as evidenced by the accumulation of multiple shunt products. While our attempt to render the $ACP_{13}/KS_{21}$ junction more native by site-directed mutagenesis did result in certain target metabolites, the most significant effect was to increase the yields of the truncated chains.

Having narrowed down the biosynthetic block to events occurring downstream of the engineered junction, we next carried out interface engineering based on PKS XUs, leveraging fusion points within the KS domain (Fig. 5). In this case, sites were selected to either maintain essentially the whole of $KS_{14}$, or to create a hybrid $KS_{14}/KS_{21}$ domain. This strategy at once preserved key elements of the $ACP_{13}/KS_{14}$ chain transfer junction, and in the case of the almost full-length $KS_{14}$, ensured that the domain

had the appropriate substrate specificity for the incoming chain. Interestingly, the construct incorporating the chimeric KS functioned best, producing the desired mini-stambomycins in both linear and macrocyclic forms. These data thus identify this location in the middle of the $KS_{53}$ as a potentially general fusion site, perhaps because it maintains key contacts with the two partner ACP domains ($ACP_{13}$ and $ACP_{21}$, in this case). Intriguingly, an analogous site within the condensation (C) domains forms the basis for the XUCs (eXchange Unit Condensation domain) of NRPS systems, which are now used routinely to generate productive hybrids[65].

Our results also showcase the intrinsically high tolerance of the Pks9 TEI domain towards shorter substrates. Indeed, the TEI domain participates in off-loading shunt metabolites from the upstream subunit, an activity which interferes with passage of the chain to subsequent modules. Unfortunately, our attempts to boost yields of the mini-stambomycins by engineering the condensation interface between $KS_{14}$ and $ACP_{21}$ were unsuccessful, both when the full $ACP_{21}$ recognition loop/helix αII region was swapped for that of $ACP_{14}$, and when a single site-directed mutation was made at a putatively critical position (Fig. 5). This result is surprising, as both of these modifications were reported in vitro to improve chain extension carried out by mismatched KS and ACP domains sourced from the erythromycin PKS (DEBS)[50]. Apparently, the introduced changes were not sufficient to ensure effective communication between the $KS_{14}$ and $ACP_{21}$ domains or were in fact deleterious to function, and/or any benefit was masked by the poor tolerance of the downstream modules to the modified intermediates.

To fully judge the efficacy of this work, it is instructive to compare it to the other two examples in the literature in which full biosynthetic systems have been re-engineered to remove multiple internal modules[12]. In the first, recently-reported case, the neoaureothin (Nor) hexamodular PKS was morphed into the evolutionarily-related aureothin (Aur) tetramodular PKS by removing the second bimodular subunit, NorA′. As in our work, the authors initially attempted to engineer an interaction between the monomodular subunits NorA and NorB flanking NorA′ using compatible docking domains, by exchanging the type 1b $^{N}DD$ of NorB for the type 1a $^{N}DD$ of NorA′ (the natural partner of NorA). When the target metabolite was not obtained, they relocated the fusion site to the KS-AT linker downstream of the conserved KS region in NorB, thereby maintaining the native NorA ACP-$^{C}DD$/$^{N}DD$-KS NorA′ junction. Ultimately, several linker variants had to be evaluated before a functional sequence was identified, in part by serendipity—indeed it is 1 residue longer than the native linker. Overall, the yields of the targeted chain-shortened metabolites dropped ~18-fold compared to the parental neoaureothin (to ca. 2.5 mg L$^{-1}$), a similar penalty as engendered by our engineering strategy. Presumably, the useful titers obtained in this experiment reflect the intrinsic amenability of the Nor PKS to conversion into an Aur PKS, as the Nor PKS likely evolved from an Aur PKS by subunit insertion[12]. Nevertheless, the newly created NorA/NorB interface was also only partially functional, as product corresponding to the intermediate generated by iterative action of the upstream subunit NorA was still obtained.

The second relevant study concerns the accelerated evolution (AE) of the RAPS PKS, based on spontaneous induced homologous recombination between its component modules[53]. As mentioned earlier, several of the resulting systems incorporated intermodular fusion sites essentially at the mid-point of the respective KS domains, and so can be compared to our best performing construct hy59_S2. Notably, yields from the hybrid RAPS PKSs from which either 3 or 6 modules were removed, were reduced by a maximum of 3-fold relative to that of the

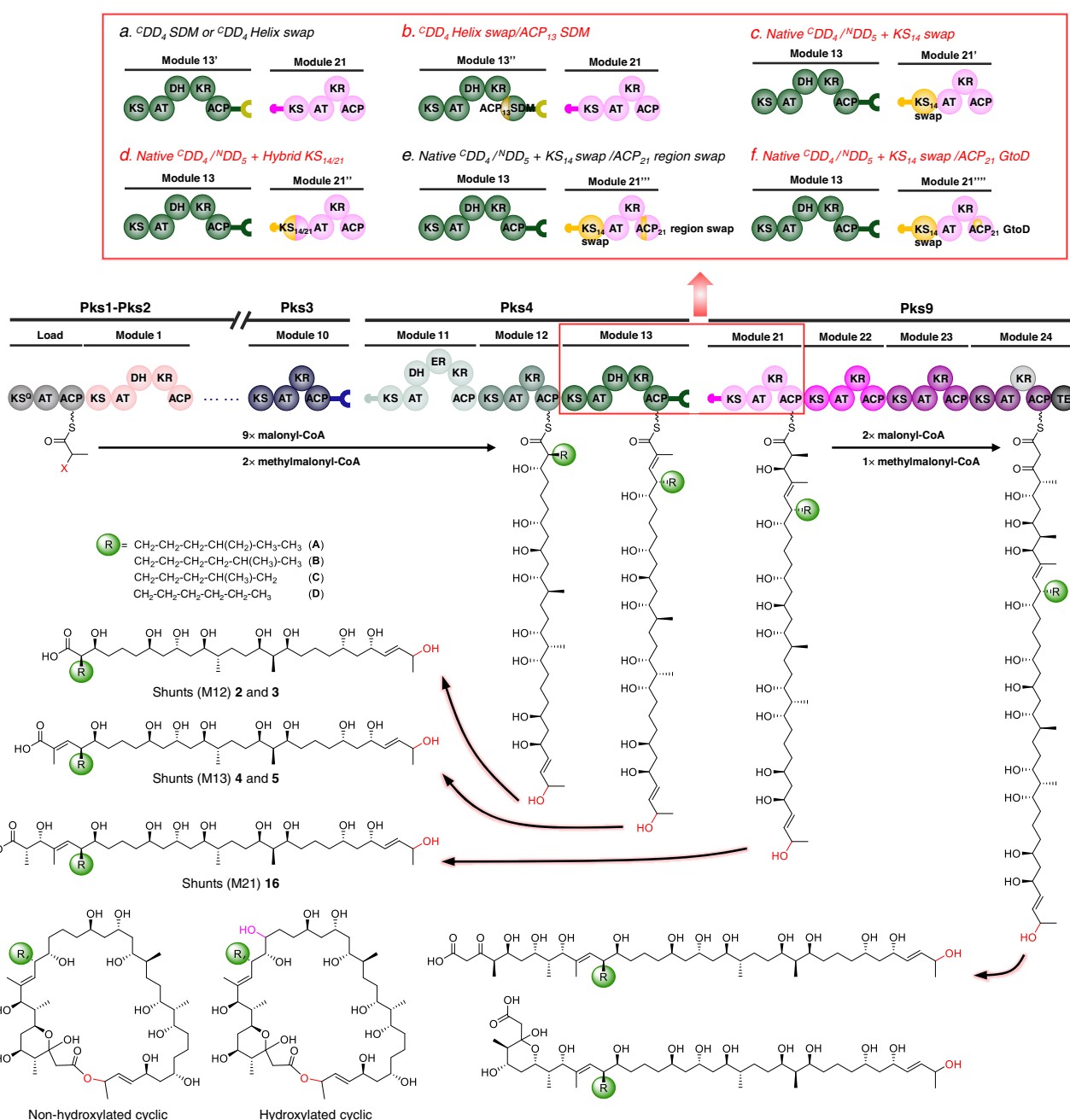

**Fig. 5 Summary of the engineering strategies applied in this work to the stambomycin PKS.** Inset (panels **a–f**) are the six distinct approaches used, and the structures of the resulting metabolites are shown. The strategies giving rise to the target mini-stambomycins **12–14** are indicated in red lettering. The hydroxyl group shown in pink is introduced by the P450-hydroxylase SamR0478, and that in red, by SamR0479. As in Figs. 1 and 3, the indicated configurations have been extrapolated from those assigned to the stambomycins **1**[77]. KS ketosynthase (KS$^Q$ refers to replacement of the active site cysteine residue by glutamine), AT acyl transferase, ACP acyl carrier protein, DH dehydratase, ER enoyl reductase, KR ketoreductase, TE thioesterase, $^C$DD C-terminal docking domain, $^N$DD N-terminal docking domain.

parental compound. We propose two explanations for the higher functionality of these systems relative to hy59_S2. First of all, in every case, the module downstream of the newly formed junctions in the contracted RAPS PKSs was internal to its respective subunit (unlike module 21 of Pks9), consistent with the idea that such modules boast intrinsically broader substrate specificity[22]. Secondly, the KS domains of the RAPS PKS exhibit unusually high mutual sequence identity (85–90%)[66]. This strong similarity means, in effect, that the same KS domain acts on a large variety of substrates of differing length and functionality, contributing to the tolerance of all modules downstream of the fusion site. In contrast, the KSs in the stambomycin PKS have substantially lower sequence identity (62–80%), and thus KSs 22–24 in hy59_S2 likely represent a specificity barrier to efficient transfer and extension of the modified intermediates.

Taken together, this set of results shows that contracting PKS assembly lines represents a viable approach to accessing truncated polyketide derivatives of variable length, including macrocycles.

Whether such systems are generated rationally or using an AE process, efficiency will likely be boosted by employing (i) PKSs whose modules (and in particular KS domains) exhibit a substantial degree of mutual sequence identity and thus intrinsically high substrate tolerance, or which can be adapted by mutagenesis to broaden their specificity[37]; and, (ii) creating junctions with downstream modules which are situated at internal positions within their subunits. The data also reinforce the idea that in cases where communication at modified interfaces occurs via noncovalent protein–protein interactions, PKS XUs which include at least a portion of the KS downstream from the docking domains, should be used to boost efficiency[12,13,20]. Finally, our work has identified an increase in TEI-mediated proof-reading provoked by such interface engineering. Elucidating the mechanism underlying this unexpected intersubunit release activity, and thus how to effectively suppress it, should be a profitable avenue for further boosting product titers.

## Methods

**Bioinformatics analysis.** To underpin the interface engineering strategy, the extremities of all the stambomycin PKS subunits were analyzed to identify the boundaries of the most C-terminal and N-terminal functional domains (ACP and KS, respectively), and thus the regions potentially containing docking domains (DDs). The resulting sequences were compared by multiple sequence alignment using Clustal Omega (https://www.ebi.ac.uk/Tools/msa/clustalo)[67] to bona fide and putative DD sequences from multiple DD classes, including those present at the DEBS 2/DEBS 3 interface (type 1a, PDB ID:1PZQ [https://doi.org/10.2210/pdb1PZQ/pdb], 1PZR [https://doi.org/10.2210/pdb1PZR/pdb][26]), and the PikAIII/PikAIV junction (type 1b, PDB ID: 3F5H [https://doi.org/10.2210/pdb3F5H/pdb][35]), to allow for type classification. To identify suitable boundaries for DD heterologous expression in *E. coli*, the secondary structure of the putative DD regions was predicted using PSIPRED 4.0 (http://bioinf.cs.ucl.ac.uk/psipred/)[46]. Analysis for potential specificity-conferring residues in the stambomycin PKS ketosynthase (KS) domains was carried out by multiple sequence alignment against model KS domains[17,24,37], using Clustal Omega[67].

**General methods.** All reagents and chemicals were obtained from Sigma–Aldrich, except the following: BD (tryptone, yeast extract, TSB powder), Thermo Fisher Scientific (Tris), VWR (glycerol, NaCl, NaNO₃), ADM, France (NutriSoy flour), and New England Biolabs (T4 DNA ligase, restriction enzymes). Oligonucleotide primers and two additional synthetic DNA fragments for CPN4 and CPN5 constructs were synthesized by Sigma–Aldrich (Supplementary Data 1). The docking domains $^NDD_9$ Val and $^NDD_9$ Met (Supplementary Table 1) were obtained as synthetic peptides from GeneCust. DNA sequencing of PCR products was performed by Sigma–Aldrich and Eurofins.

PCR reactions were performed with Taq DNA polymerase (Thermo Fisher Scientific), or Phusion High-Fidelity DNA polymerase (Thermo Fisher Scientific) when higher fidelity was required. Isolation of DNA fragments from agarose gel, purification of PCR products and extraction of plasmids were carried out using the NucleoSpin® Gel and PCR Clean-up or NucleoSpin® Plasmid DNA kits (Macherey Nagel, Hoerdt, France).

**Strains and media.** *E. coli* BL21 strains were obtained from Novagen. Unless otherwise specified, all *E. coli* strains were cultured in LB medium (yeast extract 10 g, tryptone 5 g, NaCl 10 g, distilled water up to 1 L, pH 7.0)[68] or on LB agar plates (LB medium supplemented with 20 g L⁻¹ agar) at 37 °C. *Streptomyces ambofaciens* ATCC23877 and the derived mutants were grown in TSB (TSB powder 30 g (tryptone 17 g, soy 3 g, NaCl 5 g, K₂HPO₄ 2.5 g, glucose 2.5 g), distilled water up to 1 L, pH 7.3) or on TSA plates (TSB medium supplemented with 20 g L⁻¹ agar), and sporulated on SFM[69] agar plates (NutriSoy flour 20 g, D-mannitol 20 g, agar 20 g, tap water up to 1 L) at 30 °C. All strains were maintained in 20% (*v/v*) glycerol in 2 mL Eppendorf tubes and stored at −80 °C.

For fermentation of *S. ambofaciens* ATCC23877 and its mutants, spores were streaked on TSA with appropriate antibiotics and after incubation 48 h at 30 °C, a loop of mycelium was used to inoculate 7 ml of MP5 medium (yeast extract 7 g, NaCl 5 g, NaNO₃ 1 g, glycerol 36 mL, MOPS 20.9 g, distilled water up to 1 L, pH 7.4) supplemented with selective antibiotics and sterile glass beads, followed by incubation at 30 °C and 200 rpm for 24–48 h. Finally, the seed culture was centrifuged and resuspended into 2 mL fresh MP5 before being inoculated into 50 mL MP5 medium in a 250 mL Erlenmeyer flask, and cultivated at 200 rpm and 30 °C for 4 days.

**PCR-targeting-based genetic engineering.** The following protocol applies to mutant ATCC/OE484/K7N1, but the same overall procedure was used to construct mutants K7N2/OE483, K7N2/pIB139, K7N3/OE483, K7N3/pIB139, K7N4/OE483,

K7N4/pIB139, K7N5/OE483, K7N5/pIB139, K7N6/OE483, and K7N6/pIB139. To render the BAC BAA9ZA8 proficient for selection following conjugation, its chloramphenicol resistance gene was replaced using a PCR-targeting approach, by a kanamycin resistance gene cassette sourced from pIJ776[39], resulting in BAC1 (Supplementary Data 2). The cassette *attL + aac(3)IV + oriT + attR* was amplified from the plasmid pSPM88T[70] using primers 9996 and 9997 (Supplementary Data 1), affording PCR amplicon *PCR-K7N1*. The PCR fragment was then electro-transformed into BW25113/pKD20/BAC1[71], giving rise to mutant BAC1_K7N1 (in which the C-terminus of *pks4* was modified and the genes *pks5–pks8*, were deleted)[39]. The BAC1_K7N1 was then introduced into *E. coli* ET12567/pUZ8002[72] and then transferred to *S. ambofaciens* wild-type via intergeneric conjugation. The resulting exconjugants (ATCC/K7N1_*aac(3)IV + oriT*) were selected for their apramycin resistance and kanamycin sensitivity (i.e., a phenotype consistent with successful double cross-over). The correct mutations were confirmed by PCR and sequencing. Subsequently, the disruption cassette was excised using the excisionase and integrase of pSAM2 encoded by pOSK111[70], leaving a 33 bp scar sequence (mutant ATCC/K7N1). Successful removal of the cassette was verified by PCR and DNA sequencing. Finally, the LAL regulator overexpression plasmid pOE484[33] or the parental vector pIB139[41] was introduced into the strain giving rise to mutants K7N1/OE484 and K7N1/pIB139, respectively. An analogous PCR-targeting approach was also employed to inactivate *samR0478* and *samR0479* using appropriate BACs[42] (Supplementary Data 2).

**CRISPR-Cas9-mediated genetic engineering.** Plasmids pCRISPomyces-2 (and associated cloning and screen protocols)[40] used for construction of all mutants except ATCC/hy59_S1 and ATCC/hy59_S2 and pCRISPR-Cas9[55] were used for CRISPR-Cas9-based genome editing. The two systems differ in the way in which Cas9 is expressed; in the case of pCRISPomyces-2, the nuclease is expressed constitutively, while in the pCRISPR-Cas9 system, its expression is under inductive control by thiostrepton (Tsr). The crRNA sequence was selected to match the DNA segment which contains NGG on its 3′ end (N is any nucleotide, and the NGG corresponds to the protospacer-adjacent motif (PAM)). The annealed crRNA fragment and two homologous arms (HAL and HAR, flanking the target region) were sequentially inserted into the delivery plasmid pCRISPomyces-2 using the restriction sites *Bbs*I and *Xba*I, respectively, to afford the specific recombinant plasmid for each mutant (Supplementary Fig. 6). Correspondingly, an sgRNA cassette (tracrRNA + crRNA) and two homologous arms were inserted into the plasmid pCRISPR-Cas9 using sites *Nco*I, *Sna*BI, and *Stu*I, respectively (Supplementary Fig. 31). In addition, the crRNA was designed to be located within the region to be deleted (Supplementary Fig. 6) to avoid Cas9-catalyzed cleavage occurring in the genome of the resulting mutant. In the case of site-directed mutants, additional DNA fragments containing the targeted mutations were inserted between the two homologous arms. In addition, the DNA sequence with the fragments identical to the crRNA was modified, so as to avoid subsequent Cas9-catalyzed cleavage of the obtained mutants (Supplementary Figs. 6, 28 and 39).

**Overexpression and purification of docking domains.** The wild-type docking domains ($^CDD_4$, $^NDD_5$, $^CDD_8$, $^NDD_9$ Val, and $^NDD_9$ Met) and mutant docking domains ($^CDD_4$ SDM, $^CDD_4$ helix swap) were amplified from genomic DNA of *S. ambofaciens* wild-type and the relevant mutants, using forward and reverse primers incorporating *Bam*HI and *Hin*dIII restriction sites, respectively (Supplementary Data 1). The PCR amplicons were digested using FD *Bam*HI and FD *Hin*dIII, and then ligated into the equivalent sites of vector pBG102 (Center for Structural Biology, Vanderbilt University). In the case of all $^CDD$s which lacked aromatic residues, a tyrosine residue (codon TAT incorporated in the forward primer, Supplementary Table 1) was added at the N-terminal ends (so as not to interfere with docking with the $^NDD$ partner) to allow efficient monitoring by UV-Vis during the purification, as well as reliable measurement of protein concentration necessary for binding studies by ITC.

The resulting constructs pBG102-$^NDD_5$, pBG102-$^CDD_8$, and pBG102-$^NDD_9$ were used to transform *E. coli* BL21 (DE3). For $^CDD_4$ and its mutants, these were transformed into Rosetta™ 2(DE3), as these constructs contain 8 codons rarely used in *E. coli*. Positive transformants were selected on LB agar supplemented with kanamycin (50 μg mL⁻¹) (25 μg mL⁻¹ chloramphenicol was also added for expression in Rosetta™ 2(DE3)). A single colony was transferred to LB (10 mL) supplemented with antibiotics, and the culture grown at 37 °C and 200 rpm overnight. The 1 mL overnight culture was used to inoculate LB media (1 L) supplemented with appropriate antibiotics, and then incubated at 37 °C and 200 rpm to an optical density of 0.8, at which point protein synthesis was induced by the addition of IPTG (final concentration 0.1 mM). After incubation at 18 °C and 200 rpm for 18 h, cells were collected by centrifugation at 8000 × *g* for 30 min, resuspended in 40 mL protein purification buffer A (50 mM Tris-HCl, 400 mM NaCl, 10 mM imidazole, pH 8.0), and lysed by sonication. Following centrifugation at 20,000 × *g* and filtration using a 0.45 μm membrane, the soluble cell lysates were loaded onto 2 × 5 mL HisTrap HP (GE Healthcare) columns (two 5 mL columns in series) equilibrated in buffer A, and purified by preparative protein purification chromatography using an ÄKTA Avant system. The following program was applied: sample loading, 1 mL min⁻¹; washing, 2 mL min⁻¹, 10 column volumes of buffer A; elution, 2 mL min⁻¹, 5 column volumes of buffer B (50 mM Tris-HCl,

400 mM NaCl, 250 mM imidazole, pH 8.0); elution, 2 mL min$^{-1}$, 2 column volumes of buffer C (50 mM Tris-HCl, 400 mM NaCl, 500 mM imidazole, pH 8.0).

All His$_6$-SUMO-tagged proteins were collected (fractions containing the protein of interest were selected based on the UV chromatography and SDS-PAGE gel), and transferred into dialysis bag containing His$_6$-tagged human rhinovirus 3 C protease (H3C) (1–2 µM). The dialysis bag was then placed in a container filled with buffer D (50 mM Tris-HCl, 400 mM NaCl, pH 8.0), and the cleavage allowed to proceed at 4 °C overnight. The resulting proteins, which incorporated a non-native N-terminal GPGS sequence, were then separated from the remaining His$_6$-tagged SUMO and His$_6$-tagged human rhinovirus 3 C protease by reloading onto the 2 × 5 mL HisTrap HP columns pre-equilibrated in buffer A. Purification was then carried out with the following program: sample loading, 1 mL min$^{-1}$; washing, 2 mL min$^{-1}$, 4 column volumes of buffer A; elution, 2 mL min$^{-1}$, 2 column volumes of buffer B; elution, 2 mL min$^{-1}$, 2 column volumes of buffer C. The untagged docking domains passed through the column during the washing step, and were collected and concentrated to 5–7 mL using an Amicon Ultra 3000 MWCO centrifuge filter (Millipore Corp).

Subsequently, the concentrated docking domains were loaded onto a size exclusion chromatography column (Superdex 75 26/60 column, GE Healthcare) equilibrated in buffer GF (20 mM HEPES, 100 mM NaCl, 0.5 mM TCEP, pH 7.5). Following a concentration step, the purity of the purified proteins was verified by SDS-PAGE, and their concentrations were determined by NanoDrop (Thermo Scientific) with extinction coefficients calculated using the ExPASy ProtParam tool (https://web.expasy.org/protparam/)[73].

**Isothermal titration calorimetry measurements.** Isothermal titration calorimetry (ITC) measurements were performed at 20 °C in buffer GF using a MicroCal ITC200 (Malvern Instruments) (A2F Plateforme ASIA: Approches fonctionnelles et Structurales des InterActions cellulaires). A 300 µL aliquot of $^N$DD$_5$ at 70 µM was placed in the calorimeter cell and titrated with 700 µM of the $^C$DD$_4$s ($^C$DD$_4$ wild-type, $^C$DD$_4$ SDM, and $^C$DD$_4$ helix swap) in the syringe. In the case of the binding experiments between $^N$DD$_9$ Met and $^C$DD$_8$, the $^C$DD$_8$ (700 µM) was added to the $^N$DD$_9$ Met (80 µM in the cell), while for the binding between $^N$DD$_9$ Val and the $^C$DDs ($^C$DD$_8$, $^C$DD$_4$ wild-type, $^C$DD$_4$ SDM, and $^C$DD$_4$ helix swap), the $^C$DDs (700 µM) were added to $^N$DD$_9$ Val in the cell (120 µM). The ITC experiments were then carried out as follows initial waiting time 120 s, initial injection of 0.5 µL over 1 s followed by 19 serial injections of 2 µL over 4 s, separated by an interval of 120 s. For each experiment, the reference power was set to 5 µcal$^{-1}$, stirring speed to 750 rpm, and the high feedback mode was selected. Two independent titrations were performed for each combination of DDs. The heat of reaction per injection (µcal s$^{-1}$) was determined by integration of the peak areas using the Origin 7.0 (OriginLab) software, assuming a one-site binding model (consistent with the solved structures of the types of DDs[26,35]), yielding the best-fit values for the heat of binding (ΔH), the stoichiometry of binding (N), and the dissociation constant (K$_d$). The heats of dilution of the DDs were determined by injecting them into the cell containing buffer only, and these were subtracted from the corresponding binding data prior to curve fitting.

In some cases, when a plateau (binding saturation) was not reached at the final titration step, and the problem could not be solved by increasing the concentration of DD in syringe, we initially placed $^C$DD/$^N$DD complex in the ITC cell (at the concentration of the two partners reached in the previous titration), filled the syringe with additional DD, and performed a second titration experiment. This procedure was then repeated until binding saturation was reached. To fit the data, the MicroCal Concat ITC software version 1.00 was used to combine two ITC data files together. The critical parameter dimensionless constant (C-value) was then calculated as follows:

$$C = NK_a[M]_T \qquad (1)$$

where K$_a$ is the binding constant, [M]$_T$ is the total macromolecular concentration in the cell, and N is the stoichiometry of interaction. A reliable ITC binding isotherm is evidenced by ITC data with C-values > 1 (the optimal range is 5 < C < 500)[74], as was the case for all of our measurements.

**Circular dichroism measurements.** Circular dichroism (CD) spectra were recorded on a Chirascan CD (Applied Photophysics, United Kingdom) (IBS-Lor UMS 2008 Plateforme de Biophysique et Biologie Structurale) at 0.5 nm intervals in the wavelength range of 180–260 nm at 20 °C, using a temperature-controlled chamber. A 0.01 cm quartz cuvette containing 30 µL of docking domain at 100 µM, a 0.1 cm cuvette with 200 µL of sample at 10 µM, and a 1 cm cuvette containing 1.5 mL of sample at 1 µM, were used for all the measurements. All measurements were performed at least in triplicate, and sample spectra were corrected for buffer background by subtracting the average spectrum of buffer alone. The CD spectra were deconvoluted using the deconvolution software CDNN2.1[75] to estimate the secondary structure present in the docking domains.

**SEC-MALS analysis of docking domains.** The oligomeric state of all the docking domains was determined by size exclusion chromatography multi-angle light scattering (SEC-MALS) on the A2F Plateforme ASIA. For this, SEC was first carried out on a Superdex75 10/300 column (GE Healthcare) at 20 °C using a flow rate of 0.5 mL min$^{-1}$ in HEPES buffer (20 mM HEPES, 100 mM NaCl, 0.5 mM

TCEP, pH 7.5) using an ÄKTA-Purifier FPLC (GE Healthcare). Multi-angle light scattering (MALS) was measured using a MiniDAWN TREOS II (Wyatt Technology), while refractometry was monitored using an Optilab T-rEX (Wyatt Technology). Data processing was carried out with the manufacturer-supplied software (ASTRA 6.1, Wyatt Technology) to determine the protein oligomerization state.

**LC-ESI-HRMS analysis of fermentation metabolites and purified docking domains.** The fermentation broth of *Streptomyces* was centrifuged at $4000 \times g$ for 10 min. The stambomycins and their derivatives were then extracted from the mycelia, by first resuspending the cells in 40 mL distilled water, followed by centrifugation ($4000 \times g$, 10 min, repeated 3×) to remove water-soluble components[33]. After decanting the water, the cell pellets were weighed and extracted with methanol by shaking at 150 rpm for 2 h at room temperature. Thereafter, the methanol extracts were filtered to remove the cell debris, followed by rotary evaporation to dryness. The obtained extracts were then dissolved in methanol, whose volume was determined according to the initial weight of the mycelia (70 µL methanol to 1 g of initial cell pellet). The resulting mycelial crude extracts were then passed through a 0.4 µm syringe filter and analyzed in positive electrospray mode (ESI$^+$) by HPLC-HRMS at the Université de Lorraine on either a Thermo Scientific Orbitrap LTQXL or an Orbitrap ID-X Tribrid Mass Spectrometer) (Plateau d'Analyse Structurale et Métabolomique (PASM) SF4242 EFABA) using an Alltima™ C18 column (2.1 × 150 mm, 5 µm particle size). Separation was carried out with Milli-Q water containing 0.1% formic acid (A) and acetonitrile containing 0.1% formic acid (B) using the following elution profile: 0–48 min, linear gradient 5–95% solvent B; 48–54 min, constant 95% solvent B; 54–60 min, constant 5% solvent B. Mass spectrometry operating parameters were: spray voltage, 5 kV; source gases were set respectively for sheath gas, auxiliary gas and sweep gas at 30, 10, and 10 arbitrary units min$^{-1}$; capillary temperature, 275 °C; capillary voltage, 4 V; tube lens, split lens and front lens voltages 155, −28, and −6 V, respectively. Due to the much lower sensitivity of the Orbitrap LTQXL relative to the Orbitrap ID-X Tribrid as evidenced by comparative analysis of identical samples on the two instruments, we introduced a 10× correction factor to the yields determined using the Orbitrap LTQXL (Supplementary Tables 4–7).

The purified docking domains in buffer GF were diluted with Milli-Q water to a concentration of 50 µM and injected onto an Alltima™ C18 column (2.1 × 150 mm, 5 µm particle size). Analysis was carried out with Milli-Q water containing 0.1% TFA (A) and acetonitrile containing 0.1% TFA (B), using the elution profile: 0–15 min, constant 10% solvent B; 15–20 min, linear gradient of 10% solvent B to 95%; 20–25 min, constant 10% solvent B. Mass spectrometry operating parameters were set as above.

**Metabolite profiling of engineered strains.** Comparative analysis of fermentation extracts of all strains containing pOE484 except K7N3, CPN4 and CPN5, relative to control mutant containing empty plasmid pIB139, was conducted at ETH Zurich on a Dionex Ultimate 3000 HPLC system coupled to a Thermo Scientific™ Q Exactive™ Hybrid Quadrupole-Orbitrap mass spectrometer. MS-settings: spray voltage 3.5 kV; capillary temperature 320 °C; sheath gas (52.50), auxiliary gas (13.75), sweep gas (2.75); probe heater 437.50 °C; S-Lens RF (50), positive mode, resolution 70.000; AGC target 1e6, microscans 1, maximum IT 75 ms, scan range 200–1800 *m/z*. Chromatographic separation was obtained using a Phenomenex Kinetex 2.6 µm XB-C18 150 × 4.6 mm column with solvents (A, H$_2$O + 0.1% formic acid) and (B, MeCN + 0.1% formic acid) and the following gradient: flow rate 0.7 mL min$^{-1}$, 20% B for 2 min, 20–98% B over 18 min, 98% B for 5 min, 98–20% B in 0.5 min and 20% B for 4 min. Metabolic differences within the obtained data (Supplementary Table 8) were identified using SIEVE 2.0 screening software (Thermo Fischer Scientific), applying the default settings for component extraction of small molecules, except that of the base peak minimum intensity, which was set to 5000000.

**Quantification of metabolites.** Yields of the native stambomycins were rigorously evaluated by generating a calibration curve using a previously-purified mixture of stambomycins **1**A/B[33], over the concentration range of 0.00001–0.25 mg mL$^{-1}$. This approach yielded a linear correlation between the quantity of metabolite and the respective peak area in the extracted ion chromatogram (EIC) (the areas of the peaks corresponding to the parental ions [M + H]$^+$ and [M + 2H]$^{2+}$ were used systematically) (Supplementary Fig. 9 and Supplementary Table 2).

To allow for estimating yields of the shorter, engineered metabolites that notably lacked the β-D-mycaminose of **1**, we purified 50-deoxystambomycins **2**A/B from a previously described strain of *S. ambofaciens* in which the C-50 hydroxylase had been inactivated (*S. ambofaciens* ATCC/OE484/Δ479)[42]. For this, crude extracts of the strain were fractionated using preparative reverse-phase HPLC (Agilent 1260 Infinity system, equipped with a Phenomenex Luna 5 µm C18 column (21.2 × 250 mm)). Deionized water (Milli-Q, Millipore) + 0.05% TFA (solvent A) and acetonitrile + 0.05% TFA (solvent B) were used as the mobile phase. Purification was achieved using an elution gradient of 5–100% solvent B over 50 min, followed by 100% solvent B for 10 min. Product elution was monitored by UV-Vis at 210, 254, and 280 nm. Fractions containing the 50-deoxystambomycins **2** were further purified by semi-preparative reverse-phase

HPLC (Agilent 1260 Infinity system, equipped with a Phenomenex Kinetex 5 µm C18 column (10 × 250 mm)). 50-Deoxystambomycins **2**A/B (0.85 mg) were purified using an elution gradient of 40–60% solvent B for 30 min, followed by a gradient shift from 60–100% over 5 min, and finally isocratic 100% solvent B over 5 min. The same procedure also yielded purified 50-deoxystambomycins **2**C/D (0.3 mg) and **2**F (0.3 mg). Compound identities were confirmed by HPLC-MS analysis using a Thermo Scientific™ Q Exactive™ Hybrid Quadrupole-Orbitrap mass spectrometer (ETH Zurich), as described earlier. The final masses of the purified 50-deoxystambomycins **2**A/B were determined to two decimal points of accuracy using a Mettler Toledo Excellence XS204 analytical balance.

Analysis of the 50-deoxystambomycins **2** at the same concentration range as **1** (Supplementary Fig. 11) revealed a substantially lower detection efficiency under our conditions (Orbitrap ID-X Tribrid Mass Spectrometer at the Université de Lorraine). Direct comparative analysis in triplicate of identical concentrations (0.25 mg mL$^{-1}$) of wild-type stambomycins **1**A/B and 50-deoxystambomycins **2**A/B with detection by MS using the most abundant singly- and doubly-charged ions ($[M + H]^+$ and $[M + 2H]^{2+}$ for the stambomycins **1**A/B and $[M - 3H_2O - CO_2 + H]^+$ and $[M - 3H_2O - CO_2 + 2H]^{2+}$ for the 50-deoxystambomycins **2**A/B), and by UV-Vis at 254 and 238 nm, confirmed that the sensitivity towards **1** was 206-fold higher by MS and ca. 30-fold higher by UV-Vis. We further showed that the presence of an alternative amino sugar, β-D-desosamine, in a model macrolide correlated with improved MS sensitivity, by analysis of commercial erythromycin A **3** at a range of concentrations (0.00005–0.05 mg mL$^{-1}$).

As the standard curve generated from the parental stambomycins **1**A/B spanned a 25,000-fold range of concentrations and was therefore more reliable, we used it to convert peak areas measured for all stambomycin derivatives (based on the $[M + H]^+$ and $[M + Na]^+$ ions for metabolites **4–11**, and $[M + H]^+$ for **12–17** (as no $[M + Na]^+$ peaks were present)) into yields, and then introduced a correction factor of 206 derived from the 50-deoxystambomycins **2**A/B in order to obtain overall yield range estimates for the metabolites (Supplementary Tables 4–7). Direct use of a limited calibration curve produced from the 50-deoxystambomycins **2**A/B gave similar results (Supplementary Table 3).

**Reporting summary**. Further information on research design is available in the Nature Research Reporting Summary linked to this article.

## Data availability

All data supporting the findings of this study are available within the manuscript. The docking domain structures used in the DD analysis are available in the PDB under the following accession codes: 1PZQ, 1PZR (type 1a), and 3F5H (type 1b). The raw HPLC-MS data have been deposited in the data repository DOREL (Données de la Recherche Lorraines) [https://doi.org/10.12763/PEYXHP]. Source data are provided with this paper.

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

## Acknowledgements

We acknowledge financial support from the Agence Nationale de la Recherche (grant number ANR-16-CE92-0006-01, PKS STRUCTURE to K.J.W.), the Université de Lorraine, the Centre National de la Recherche Scientifique (CNRS), and the IMPACT Biomolecules project of the Lorraine Université d'Excellence (Investissements d'avenir—ANR 15-004 to L.S., C.J., B.A., and K.J.W.). J.P. acknowledges funding by the European Research Council (ERC) under the European Union's Horizon 2020 Research and Innovation Program (grant agreement No. 742739). S. Collin and B. Chagot are thanked for help with the analysis of the DDs in vitro, A. Kriznik for assistance with the biophysical experiments, J. Grosjean for help with the HPLC-MS analysis, S. Rousselot for contributing to development of the genetic engineering strategy, and J.-M. Girardet and T. Dhalleine of the Functional and Structural Approaches to Cellular InterActions platform (ASIA, University of Lorraine-INRAE; https://a2f.univ-lorraine.fr/en/asia-2/), for help with the ITC and SEC-MALS, respectively. We are also grateful for access to the Orbitrap ID-X Tribrid system of the Structural and Metabolomics Analyses Platform for the LC-MS/MS analyses (PASM, SF4242, Université de Lorraine, EFABA, Vandoeuvre-lès-Nancy, France).

## Author contributions

L.S. constructed plasmids, generated and fermented recombinant strains, analyzed and interpreted HPLC-MS data, expressed and purified recombinant docking domains, carried out biophysical analysis of DD interactions, and generated all of the manuscript figures. L.H. and C.J. constructed certain plasmids, mutant BACs and/or mutant strains. C.P. performed LC-ESI-HRMS analyses. A.B. and J.P. carried out HPLC-MS analyses and metabolic profiling. C.C. purified the 50-deoxystambomycins. B.A., C.J., and K.J.W. designed the research and supervised the project. K.J.W. performed in silico analyses and wrote the manuscript, with contributions from all authors.

## Competing interests

The authors declare no competing interests.
