## [Peer Review File · Nature Communications]

Engineering the stambomycin modular polyketide synthase yields 37-membered mini-stambomycinsReviewers' Comments:

Reviewer #1:

Remarks to the Author:

In this effort to engineer the stambomycin biosynthetic pathway, the authors attempt to tackle a "most substantial modification to an intact system". Removal of PKS genes 5-8 reduced the final natural product from a 51 to 37-membered metabolite. The work is rigorous, and the results are noteworthy. The paper is suitable for publication pending revision to address the following:

- 1) Results sections 1 and 2 could be combined, seems repetitive to have both titles: "Design of engineering experiments based on classical modular boundaries" & "Engineering the stambomycin PKS based on the classical module definition"
- 2) The acronym SDM is not defined in the main manuscript.
- 3) The purpose of the mutants/strains (Page 4) (PCR targeting and CRISPR-Cas9) does not appear to have been explained.
- 4) Page 5; "Taken together these data show that the unusual on-line modification catalyzed by SamR0479 which is necessary for macrocyclization, occurs prior to chain extension by Pks5". Not sure there is evidence for this. The figures are not convincing regarding this.
- 5) Figure 1. The polyketide units in the final product could be color-coded to match the module from which they came.
- 6) Figure 2. The arrows legend is cut off the page.
- 7) Figure 5. Please change colors on molecules 11 and 12, very difficult to discern the difference.

Reviewer #2:

Remarks to the Author:

In this manuscript, Su et al. describe an impressive collection of bioengineering experiments with the stambomycin polyketide synthase system. The aim of these experiments was to produce 'mini' stambomycins by deleting seven internal modules (m14-20). The focus of the bioengineering work addressed a key pinch point in the engineered assembly line -- poor chain transfer from Pks4-module 13 to Pks9-module 21. The authors provide an exquisite description of how they engineered the associated docking, ACP and KS domains in their pursuit of generating a functional assembly line. While studies have achieved similar results with other polyketide systems, this work pushes the field away from blind luck and serendipity toward something that resembles rational engineering. I am inclined to agree with the authors claim in the abstract this work is the most substantial bioengineering study to date and I am confident that when this manuscript is ultimately published that it will be received by the community with open arms.

I do not have editorial suggestions - the manuscript itself is very well crafted and illustrated. I have one relatively minor experimental comment. I appreciate and sympathize that the low compound titer has precluded NMR structural confirmation of shunts and mini-stambomycins. I agree that production of the entire 'suite' of mini-compounds in both the wild-type and Δ Sam0479 is good evidence that the compounds originate from the engineered pathway, how a slightly more rigorous mass spectrometry analysis is warranted than what is shown in Supp Figure 10C. For instance, in Supp Figure 5, the authors show detailed mass characterization of wild-type stambomycins and shunts. The story would be stronger if the same was shown in Supp Figure 10C for mini-stambomycins. I suspect this data has already been collected and addressing this issue would involve some re-analysis of the data.

Reviewer #3:

Remarks to the Author:

The manuscript by Su et al. focuses on efforts to manipulate the stambomycin modular polyketide synthase with the goal of generating new macrolactones and macrolide molecules. They employ

genetic methods in the stambomycin producing microorganism (*Streptomyces ambofaciens* ATCC23877) to alter the ring size of the 51 membered macrolactone to a series of smaller rings. Their approach involved deletion of a series of modules encompassing the polymodular Pks5 – Pks8 region of the gene cluster. In order for the new, unnatural interaction between Pks4 and Pks9 to be optimized, the authors sought to engineer the new interface (between these two proteins at the carboxy terminus of Pks4 and the amino terminus of Pks9). PCR-targeting and CRISPR-Cas9 based methods were employed to generate different constructs for interface optimization, and these unnatural engineered PKS proteins were tested for their ability to generate the predicted 37-membered ring macrolactone core molecules. Overall, the authors experienced similar challenges and difficulties that have been reported in conceptually similar experiments going back to studies on the erythromycin system in the mid-1990s and 2000s by Khosla, Santi and colleagues (citations #8, #9, #12). Specifically, no matter how sophisticated the genetic methods, and efforts to optimize interfaces for module::module interaction between unnatural PKS proteins, the efficiency of the PKS biochemical machine is severely attenuated and the production of new molecules becomes vanishingly small. The current paper reinforces this point in a system with so many variables and complexities that it is difficult to adhere to a single focused take home other than the following. These authors did a tremendous amount of work on an interesting system, but what have we learned that represents a contribution others in the field can apply in the future?

Major concerns.

1) Su et al. tests two current hypotheses (classical model and alternative model (Fig. 2) regarding optimization of the PKS interfaces. The authors state that novel metabolites were observed using both methods for optimization. However, which method is better remains unclear and depends on numerous parameters that were acknowledged, but largely remain undefined (see further below) in the stambomycin system. Figure 4 of the manuscript reports yields of the new metabolites that range from 0.35 to 677 micrograms/L. Over half of these metabolites are produced in such low quantities that massive scale-up (1000 fold) would be required for sufficient material to obtain a full NMR dataset.

2) Herein lies another major concern. Specifically, the authors rely solely on high resolution mass spectrometry for yield determination (Figure 4, column reporting production in micrograms/L) and structural assignments, including stereochemistry at chiral centers. Yields are based on comparison to the known wildtype stambomycin standard and its ionization the MS. The flaw in this approach is that it is not possible to accurately quantify yields using mass spectrometry unless each molecule can be compared to its own authentic standard using a standard curve. This is due to the very likely scenario that different structures have different inherent ionization parameters, which are reflected in individual peak intensities. Unless these standards exist, it is not possible to quantify production based on MS alone. Moreover, inference of structures of new compounds cannot be accomplished by mass spectrometry. If the work involved a very minor set of structural changes this might be acceptable, but the current work involves significant structural alteration, and the LC-MS analysis can provide ambiguous data with isomeric molecules generating identical mass spectra, precluding definitive structural assignment (see Koch et al. 2020, doi.org/10.1002/anie.202004991 for an example of this pitfall). For progress to be made in this very challenging domain of PKS engineering, it is critical to obtain full MS and NMR datasets to establish the true identity of the molecules and uncover any additional unexpected perturbation in PKS function that may result from this type of engineering effort.

3) In addition to engineering the docking domains for attempted optimization of the unnatural Pks4::PKS9 interaction and resulting pathway in the stambomycin system, the authors recognized other parameters at play. Specifically, the nature of the downstream KS domain is likely involved, and some efforts were made to optimize in Pks9. However, much more work is required for new insights to be gained and these efforts were not a significant aspect of the current manuscript. Likewise, the authors conducted some work on the TE domain and TEII. This domain has been recognized by others as a significant bottleneck, perhaps a primary issue in maximizing metabolic output and product yields

(Boddy et al., doi: 10.1039/c4np00148f, Koch et al., doi: 10.1021/jacs.7b06432). However, this issue was only minimally addressed in the current manuscript. Indeed, the purported identification of both cyclic and linear products (although their true identity remains unclear due to reliance on MS alone) speaks to the well-known sensitivity of the TE domain, which is certainly going to be a major issue in accommodating unnatural, shortened polyketide chains in the stambomycin system.

Reviewer #1:

In this effort to engineer the stambomycin biosynthetic pathway, the authors attempt to tackle a “most substantial modification to an intact system”. Removal of PKS genes 5-8 reduced the final natural product from a 51 to 37-membered metabolite. The work is rigorous, and the results are noteworthy. The paper is suitable for publication pending revision to address the following:

Response: We thank the reviewer for this enthusiastic evaluation of our work.

1) Results sections 1 and 2 could be combined, seems repetitive to have both titles: “Design of engineering experiments based on classical modular boundaries” & “Engineering the stambomycin PKS based on the classical module definition”

Response: As suggested, these two sections have been combined into one entitled: “Design of engineering experiments based on classical modular boundaries”.

2) The acronym SDM is not defined in the main manuscript.

Response: A definition has been provided: site-directed mutagenesis

3) The purpose of the mutants/strains (Page 4) (PCR targeting and CRISPR-Cas9) does not appear to have been explained.

Response: In the original manuscript we stated that by using both PCR targeting and CRISPR-Cas9, we aimed ‘to directly compare the efficacy of these two approaches, as well as evaluate the effect of the short scar sequence remaining in the chromosome following PCR-targeting.’ As we believe this to be a clear rationale for the mutant strains, we have not modified this section of the text.

4) Page 5; “Taken together these data show that the unusual on-line modification catalyzed by SamR0479 which is necessary for macrocyclization, occurs prior to chain extension by Pks5”. Not sure there is evidence for this. The figures are not convincing regarding this.

Response: We would like to re-argue that the evidence for the timing is convincing. Specifically, in the presence of wild type SamR0479, the metabolites released prematurely from Pks4 (compounds 4–7) (*note new compound numbering*) incorporate the C-terminal hydroxyl, but when the enzyme is inactivated by mutagenesis, the hydroxyl disappears from the chains generated by the same modules (resulting in compounds 8–11). This observation is consistent with the hydroxylase acting at some point during chain assembly upstream of and including Pks4.

5) Figure 1. The polyketide units in the final product could be color-coded to match the module from which they came.

Response: We thank the reviewer for this constructive suggestion, and the modification has been made.

6) Figure 2. The arrows legend is cut off the page.

Response: We thank the referee for noticing this error, which has now been corrected.

7) Figure 5. Please change colors on molecules 11 and 12, very difficult to discern the difference.

Response: The suggested modification, which indeed did improve the clarity of the figure, has been made.

Reviewer #2:

In this manuscript, Su *et al.* describe an impressive collection of bioengineering experiments with the stambomycin polyketide synthase system. The aim of these experiments was to produce ‘mini’ stambomycins by deleting seven internal modules (m14-20). The focus of the bioengineering work addressed a key pinch point in the engineered assembly line -- poor chain transfer from Pks4-module 13 to Pks9-module 21. The authors provide an exquisite description of how they engineered the associated docking, ACP and KS domains

in their pursuit of generating a functional assembly line. While studies have achieved similar results with other polyketide systems, this work pushes the field away from blind luck and serendipity toward something that resembles rational engineering. I am inclined to agree with the authors claim in the abstract this work is the most substantial bioengineering study to date and I am confident that when this manuscript is ultimately published that it will be received by the community with open arms.

Response: We are delighted by the reviewer's positive assessment of our work.

I do not have editorial suggestions - the manuscript itself is very well crafted and illustrated. I have one relatively minor experimental comment.

I appreciate and sympathize that the low compound titer has precluded NMR structural confirmation of shunts and mini-stambomycins. I agree that production of the entire 'suite' of mini-compounds in both the wild-type and Δ Sam0479 is good evidence that the compounds originate from the engineered pathway, how a slightly more rigorous mass spectrometry analysis is warranted than what is shown in Supp Figure 10C. For instance, in Supp Figure 5, the authors show detailed mass characterization of wild-type stambomycins and shunts. The story would be stronger if the same was shown in Supp Figure 10C for mini-stambomycins. I suspect this data has already been collected and addressing this issue would involve some re-analysis of the data.

Response: As requested by this reviewer and by reviewer #3, we have substantially boosted our MS/MS-based analysis of all novel compounds (with the exception of **12**, which was produced only intermittently and in the lowest yields of all metabolites). The obtained data are entirely consistent with each other, and with our originally-proposed structures (please see response to reviewer #3 for complete details).

Reviewer #3:

The manuscript by Su *et al.* focuses on efforts to manipulate the stambomycin modular polyketide synthase with the goal of generating new macrolactones and macrolide molecules. They employ genetic methods in the stambomycin producing microorganism (*Streptomyces ambofaciens* ATCC23877) to alter the ring size of the 51 membered macrolactone to a series of smaller rings. Their approach involved deletion of a series of modules encompassing the polymodular Pks5–Pks8 region of the gene cluster. In order for the new, unnatural interaction between Pks4 and Pks9 to be optimized, the authors sought to engineer the new interface (between these two proteins at the carboxy terminus of Pks4 and the amino terminus of Pks9). PCR-targeting and CRISPR-Cas9 based methods were employed to generate different constructs for interface optimization, and these unnatural engineered PKS proteins were tested for their ability to generate the predicted 37-membered ring macrolactone core molecules.

Overall, the authors experienced similar challenges and difficulties that have been reported in conceptually similar experiments going back to studies on the erythromycin system in the mid-1990s and 2000s by Khosla, Santi and colleagues (citations #8, #9, #12). Specifically, no matter how sophisticated the genetic methods, and efforts to optimize interfaces for module::module interaction between unnatural PKS proteins, the efficiency of the PKS biochemical machine is severely attenuated and the production of new molecules becomes vanishingly small. The current paper reinforces this point in a system with so many variables and complexities that it is difficult to adhere to a single focused take home other than the following. These authors did a tremendous amount of work on an interesting system, but what have we learned that represents a contribution others in the field can apply in the future?

Response: While it is true that rational, module-based engineering has been reported previously, this study showcases the first successful attempt, to our knowledge, of using this approach to truncate a PKS by removal of multiple internal subunits.

As a result of this reviewer's valid concerns concerning the reliability of metabolite quantification by MS (and see further comments below), we re-evaluated our quantification strategy for the targeted mini-stambomycins (see **Supplementary Fig. 6**). Specifically, in addition to the previously generated standard curve using the parental macrocyclic stambomycins **1A/B**, we newly purified and analyzed 50-deoxystambomycins

2A/B (previously reported in doi: 10.1038/ja.2013.119) by HPLC-MS (resulting in addition of another author, C Chepkirui) to the paper). In common with the majority of our engineered derivatives, these compounds are linear carboxylic acids, and critically, lack the β -D-mycaminose moiety of the stambomycins **1** which contains a protonatable nitrogen which favors ionization. Indeed, comparative analysis of the parental **1** and 50-deoxystambomycins **2** revealed that, under our analysis conditions, the detection efficiency for the 50-deoxystambomycins **2** by MS was substantially lower than for the parental compounds (factor of 206). Although we still cannot claim to have rigorously measured the mini-stambomycin yields, the clear implication here is that we almost certainly underestimated the titers of the derivatives in our original calculations.

In the revised manuscript, we have now introduced a correction factor derived from this comparison, allowing us to propose yield *ranges* (not precise values) for all of shunt metabolites and the mini-stambomycins. In each case, the lower extreme is based upon direct extrapolation from quantification of the stambomycins **1A/B**, while the upper extreme derives from the 50-deoxystambomycins **2A/B** correction factor. As the 50-deoxystambomycins **2** are arguably more faithful quantification surrogates for the range of derivatives generated in this work than the parental compounds **1**, the newly-calculated titers now argue in favor of the overall efficacy of our engineering strategy. These considerations are discussed in the revised text (pages 4–5).

Major concerns.

1) Su et al. tests two current hypotheses (classical model and alternative model (Fig. 2) regarding optimization of the PKS interfaces. The authors state that novel metabolites were observed using both methods for optimization. However, which method is better remains unclear and depends on numerous parameters that were acknowledged, but largely remain undefined (see further below) in the stambomycin system. Figure 4 of the manuscript reports yields of the new metabolites that range from 0.35 to 677 micrograms/L. Over half of these metabolites are produced in such low quantities that massive scale-up (1000 fold) would be required for sufficient material to obtain a full NMR dataset.

Response: While it is true that both of the overall strategies resulted in at least one mini-stambomycin, even the originally reported yields attested to the superiority of the alternative (now referred to as XU) approach. Specifically, while the XU strategy resulted in the full range of target metabolites **12–14**, the classical strategy only yielded detectable **13**, the most abundant of the three compounds. This conclusion is reiterated in the discussion section of the paper (page 12), where we propose a set of recommendations for future module engineering experiments.

As to the low yields, as discussed previously, we have revised our titer estimates upwards. In this context, it might in principle be possible to purify shunt metabolites **4–11**. However, the interest here is minimal, as our combined MS/MS data are already fully consistent with the proposed structures. Furthermore, no modifications were made to the PKS domains responsible for establishing the structures of these compounds, and thus there is no reasonable expectation that the polyketide chains should differ from those of the wild type intermediates.

In the case of the target mini-stambomycins (**13** and **14**), while the yield corrections indicate that the titers are *relatively* higher than initially appreciated, they remain poor in *absolute* terms because those of even the parental stambomycins **1** are low (max. 30 mg L⁻¹). This observation, coupled with their poor sensitivity of detection by MS and unreliable production (as seen in **Supplementary Table 5**), means that it would indeed require massive scale-up to obtain the quantities of material necessary for complete structure elucidation by NMR. Instead, we elected to also carry out detailed structural analysis by MS/MS of these compounds, providing a large set of internally-consistent data which again fully support the proposed structures.

2) Herein lies another major concern. Specifically, the authors rely solely on high resolution mass spectrometry for yield determination (Figure 4, column reporting production in micrograms/L) and structural assignments, including stereochemistry at chiral centers. Yields are based on comparison to the known wildtype stambomycin standard and its ionization the MS. The flaw in this approach is that it is not possible to accurately quantify yields using mass spectrometry unless each molecule can be compared to its own authentic standard using a standard curve. This is due to the very likely scenario that different structures have different inherent

ionization parameters, which are reflected in individual peak intensities. Unless these standards exist, it is not possible to quantify production based on MS alone. Moreover, inference of structures of new compounds cannot be accomplished by mass spectrometry. If the work involved a very minor set of structural changes this might be acceptable, but the current work involves significant structural alteration, and the LC-MS analysis can provide ambiguous data with isomeric molecules generating identical mass spectra, precluding definitive structural assignment (see Koch *et al.* 2020, doi.org/10.1002/anie.202004991 for an example of this pitfall). For progress to be made in this very challenging domain of PKS engineering, it is critical to obtain full MS and NMR datasets to establish the true identity of the molecules and uncover any additional unexpected perturbation in PKS function that may result from this type of engineering effort.

Response: For the comments about the yields, please see the earlier response. Concerning the stereochemistry, we must disagree with the reviewer, as we did not attempt to use mass spectrometry to assign stereochemistry. Instead, we extrapolated stereochemistries for the novel metabolites from the proposed structures of the stambomycins **1**, whose configurations were initially *predicted* based on the analysis of known ketoreductase domain stereochemical determinants. Notably, these stereochemical assignments have recently been experimentally validated by total synthesis for the C-1–C-27 fragment of the stambomycins **1** (<https://doi.org/10.1021/acs.orglett.1c02650>). We have now made these considerations clearer in the legends to **Figs. 3** and **5**.

We agree with the reviewer that, *in general*, it is not possible to infer the structures of new compounds by mass spectrometry. However, in this particular case, all of the products are derivatives of known structures which are generated by PKS multienzymes with predictable behavior. Indeed, only subunit Pks9 acts subsequent to the point of engineering, and importantly, there are only a limited number of ways in which it might function aberrantly to generate unexpected products. Specifically, the component modules 21–23 contain only KR processing domains in addition to the minimal KS, AT and ACP domains required for chain extension, while the KR of module 24 is inactive. Thus, the only perturbations possible for modules 21–23 would be for the KR domains to fail to act. This result can be definitively excluded, however, for all but metabolites **15** and **17**, based on the strong agreement between the calculated and observed accurate masses (see **Supplementary Fig. 5a**), as such behavior would result in a –2 Da change in mass. We additionally present new MS/MS data obtained on **16**, the advanced intermediate released from module 21, and its deoxy derivative **17** (see **Supplementary Fig. 11g**), which fully support their proposed structures. While it remains theoretically possible that one of modules 21–23 could act iteratively while another fails to act – a mechanism that could result in compounds of identical mass, though not of the same stereochemistry – the simplest explanation for the observed masses is that all of the modules of Pks9 have functioned normally and in the expected sequence (module 24 must always act, as it alone results in no ketoreduction).

Interestingly, in the case of metabolites **15** and **17**, we did in fact observe peaks in the MS spectra corresponding to –2 Da from the predicted masses, which are consistent with the presence of ketone groups instead of the expected hydroxyl moieties, and thus with the absence of ketoreduction in certain chain extension cycles (see **Supplementary Fig. 11g**). Therefore, when PKS functional perturbations do occur, they can be detected by our methods.

The reference cited by the reviewer is relevant to the chain liberation portion of the biosynthesis catalyzed by Pks9 module 24. At this stage, the thioesterase (TE) domain could also act aberrantly due to encountering non-native substrates (as described in Koch *et al.* 2020), and/or spontaneous chemistry could occur. To address these possibilities, we have predicted the structures of all the compounds which would result from the various alternative modes of chain release (see **Supplementary Fig. 11d**). Based on detailed MS/MS analysis, we now present evidence which makes each of these alternative less likely than the formation of macrocyclic structures **13** and **14** (**Supplementary Fig. 11d**). Additional critical support for this mode of release, and thus the proposed structures of **13** and **14** at least, is provided by inactivation of the P450 SamR0479 which is responsible for installing the hydroxyl group employed in macrocyclization. The resulting mutant strain no longer produces compounds **12–14**, but new metabolite **15** whose exact mass and structure are fully consistent with linear, deoxy mini-stambomycins (see new data in **Supplementary Fig. 11g**) – exactly the result expected for removal of the macrocyclization nucleophile, and inconsistent with *any* of the other potential release mechanisms.

Thus overall, while we were not able to obtain NMR datasets for the engineered metabolites, **the weight of evidence (as detailed below) fully supports the proposed structures of 4–17** (for **12**, the low yields precluded characterization beyond its accurate mass).

1. None are present in the control extracts
2. The strong agreement between the calculated and observed exact masses is consistent with the expected atomic compositions. Critically, there is limited alternative enzymology possible with Pks9 and/or spontaneous chemistry that would lead to the same masses for the advanced intermediates/products **12–17**.
3. For each analogue, as predicted, A/B and C/D variants are present, as evidenced by their exact masses and fragmentation patterns. The A/B and C/D variants show, in addition, consistent differences in retention time (ca. 1 min). Together, these data demonstrate that **4–17** are stambomycin derivatives.
4. For linear metabolites **4–11**, the MS fragmentation data allow for counting the number of OH groups in the molecules, and also directly attest to the presence of a terminal CO₂.
5. MS² data on **4–11** and **13–17** (and **2**) reveal a common, extensive fragment series, demonstrating the presence of shared structure.
6. All linear metabolites **4–11** and **15–17** (and **2**) give rise to a diagnostic MS² fragment of 173.1, while the macrocyclic metabolites **13** and **14** (and **1**) produce a similarly characteristic fragment at 174.1.
7. Inactivation of the P450 hydroxylase SamR0479 leads in each case to the shift in metabolic distribution precisely expected for the absence of the terminal OH: **4–7** → **8–11**; **12–14** → **15**; **16** → **17**. These shifts occur together for strains producing multiple compound series, as expected (i.e. no strain produces both metabolites bearing hydroxyls and the equivalent deoxy compounds (**Table 1**)). Critically in the case of the macrocyclic compounds **13** and **14**, this shift helps to exclude alternative modes of release than macrocyclization.

3) In addition to engineering the docking domains for attempted optimization of the unnatural Pks4:KS9 interaction and resulting pathway in the stambomycin system, the authors recognized other parameters at play. Specifically, the nature of the downstream KS domain is likely involved, and some efforts were made to optimize in Pks9. However, much more work is required for new insights to be gained and these efforts were not a significant aspect of the current manuscript. Likewise, the authors conducted some work on the TE domain and TEII. This domain has been recognized by others as a significant bottleneck, perhaps a primary issue in maximizing metabolic output and product yields (Boddy et al., doi: 10.1039/c4np00148f, Koch et al., doi: 10.1021/jacs.7b06432). However, this issue was only minimally addressed in the current manuscript. Indeed, the purported identification of both cyclic and linear products (although their true identity remains unclear due to reliance on MS alone) speaks to the well-known sensitivity of the TE domain, which is certainly going to be a major issue in accommodating unnatural, shortened polyketide chains in the stambomycin system.

Response: We agree with the reviewer that further experiments directed at the KS domains of Pks9 would be informative, and could potentially further boost the yields of the mini-stambomycins. However, given the extensive engineering already carried out on the system, we consider these experiments to be beyond the scope of the present ms. Concerning the TEI domain, given the evidence provided that both **13** and **14** are macrocyclic, and their relative high abundance compared to the linear mini-stambomycin **12** (specifically, 73% of the mini-stambomycins generated by strain ATCC/OE484/hy59_S2 were macrocyclic, and 78% of those arising from ATCC/OE484/hy59_S1 (see page 9)), the TEI would rather appear to have usefully broad substrate tolerance.

We hope that the revised manuscript is now suitable for publication in *Nat. Commun.*, and look forward to hearing from you in due course.

Yours sincerely,

Kira J. Weissman (with B Aigle and C Jacob)

Molecular and Structural Enzymology Group
UMR 7365 CNRS-UL:IMoPA
Lorraine University
Faculté de médecine, Batiment Biopôle
9 avenue de la Forêt de Haye, BP 184
54506 Vandoeuvre-Lès-Nancy - FRANCE
kira.weissman@univ-lorraine.fr
Tel : +33 3 83 68 55 44
Fax : +33 3 83 68 55 09

Contents of revised submission:

Manuscript, including 5 figures and 1 table
Supplementary information comprising 12 supplementary figures, 6 supplementary tables and supplementary references
Editorial policy checklist
Reporting summary
Source data file (Excel)

Reviewers' Comments:

Reviewer #1:

Remarks to the Author:

Accept

Reviewer #2:

Remarks to the Author:

The authors have satisfactorily addressed Reviewer #2's previous request for more detailed MS2 characterisation of mini-stambomycins. I have no further comments, other than to thank the authors for producing such an exciting study.

Response to reviewers, NCOMMS-21-04988A

There were no reviewer comments in this round to which we needed to respond. Their responses are reproduced below:

REVIEWERS' COMMENTS

Reviewer #1 (Remarks to the Author):

Accept

Reviewer #2 (Remarks to the Author):

The authors have satisfactorily addressed Reviewer #2's previous request for more detailed MS2 characterisation of mini-stambomycins. I have no further comments, other than to thank the authors for producing such an exciting study.

[Editor: Reviewer #3 is unavailable. Reviewer #2 checked the responses and states that (s)he thinks Reviewer #3's previous suggestions have been addressed in Remark to Editor section.]